# A new urban surface model integrated in the large-eddy simulation model PALM (PALM-USM v1.0)

Jaroslav Resler[1, 2], Pavel Krč[1, 2], Michal Belda[1, 2, 4], Pavel Juruš[1, 2], Nina Benešová[1, 3], Jan Lopata[1, 3], Ondřej Vlček[1, 3], Daša Damašková[1, 3], Kryštof Eben[1, 2], Přemysl Derbek[1], Björn Maronga[5], and Farah Kanani-Sühring[5]

[1]Faculty of Transportation Sciences, Czech Technical University in Prague, Czech Republic
[2]Institute of Computer Science, The Czech Academy of Sciences, Prague, Czech Republic
[3]Air Quality Protection Division, Czech Hydrometeorological Institute, Prague, Czech Republic
[4]Department of Atmospheric Physics, Faculty of Mathematics and Physics, Charles University, Prague, Czech Republic
[5]Institute of Meteorology and Climatology, Leibniz Universität Hannover, Hannover, Germany

*Correspondence to:* Jaroslav Resler (reslejar@fd.cvut.cz)

**Abstract.** Urban areas are an important part of the climate system and many aspects of urban climate have direct effects on human health and living conditions. This implies that reliable tools for local urban climate studies supporting sustainable urban planning are needed. However, a realistic implementation of urban canopy processes still poses a serious challenge for weather and climate modelling for the current generation of numerical models. To address this demand, a new Urban Surface Model (USM), describing the surface energy processes for urban environments, was developed and integrated as a module into the large-eddy simulation model PALM. The development of the presented first version of USM originated from modelling the urban heat island during summer heat wave episodes and thus implements primarily processes important in such conditions. The USM contains a multi-reflection radiation model for short- and longwave radiation with an integrated model of absorption of radiation by resolved plant canopy (i.e. trees, shrubs). Furthermore, it consists of an energy balance solver for horizontal and vertical impervious surfaces, thermal diffusion in ground, wall, and roof materials, and it includes a simple model for the consideration of anthropogenic heat sources. The USM was parallelized using the standard Message Passing Interface and performance testing demonstrates that the computational costs of the USM are reasonable on typical cluster configurations for the tested configurations. The module was fully integrated into PALM and is available via its online repository under GNU General Public License (GPL). The USM model was tested on a summer heat wave episode for a selected Prague crossroads. The general representation of the urban boundary layer and patterns of surface temperatures of various surface types (walls, pavement) are in good agreement with in-situ observations made in Prague. Additional simulations were performed in order to assess the sensitivity of the results to uncertainties in the material parameters, the domain size, and the general effect of the USM itself. The first version of the USM is limited to the processes most relevant to the study of summer heat wave and serves as a base for ongoing development which will address additional processes of the urban environment and lead to improvements to extend the utilization of USM to other environments and conditions.

# 1 Introduction

## 1.1 Urban climate

As more than half of the human population resides in cities, and this figure is expected to keep increasing in future (United Nations 2014), the influence of urban surfaces on the local urban climate gains more importance. Many aspects of urban climate have direct effects on human health and living conditions, the most prominent examples being thermal comfort and air quality.

One major phenomenon related to the urban climate is the urban heat island (UHI), i.e. the fact that an urban area may be significantly warmer than its surrounding rural areas, which mainly appears during evening and early-night hours (Oke, 1982). The higher temperature is linked to the absorption and retention of energy by urban surfaces and to anthropogenic heat emissions, which can cause urban-to-rural temperature differences of several degrees Celsius. Moreover, buildings and other urban components can locally decrease the ventilation (e.g. Letzel et al., 2012), thus adding to thermal discomfort. Chemical processes, and consequently air quality, are also affected by the urban environment.

Effects of the urban heat island on living conditions have been in the focus of urban planning for several decades in various cities, as it is anticipated that careful planning can alleviate some of these effects. However, developing adaptation and mitigation strategies requires state-of-the-art tools applicable for urban climatology studies. The work presented in this paper started in the larger framework of the project Urban Adapt[1], which focused on the development of such strategies for three major cities in the Czech Republic (Prague, Brno and Pilsen). The aim was to provide as detailed description of the street canyon conditions as possible, going to the resolution in the order of a few metres. Below we provide a brief description of the methods typically used for such a task and the motivation for developing a new Urban Surface Model (USM).

Several possible approaches for studying urban climate have been used, ranging from observation analyses, over physical modelling to numerical simulations (for a comprehensive review see e.g. Mirzaei and Haghighat, 2010; Moonen et al., 2012). In this context, a number of physical processes and their complex interactions must be taken into account (e.g. Arnfield, 2003). Urban surfaces are affected by shortwave and longwave radiation, and energy is exchanged between the urban canopy and the atmosphere in various forms, including sensible and latent heat fluxes. These fluxes in turn, together with boundary layer processes and large-scale synoptic conditions, affect the turbulent flow of air. The complexity is further increased by the presence of vegetation and the pronounced heterogeneity of urban surface materials.

For numerical modelling of urban climate processes, various models and frameworks have been used (Mirzaei and Haghighat, 2010; Moonen et al., 2012; Mirzaei, 2015). One possible approach is to use a regional meteorological or climate model. However, these models typically operate with horizontal resolutions in the order of hundreds of metres to tens of kilometres, and urban processes are treated using bulk parameterizations or single/multi-layer urban canopy models (e.g. Kusaka et al., 2001; Martilli et al., 2002). Thus, these models are much better suited to assess the influence of urban environment on the larger scale meteorology.

A second approach is represented by standalone parameterized models, e.g. the SOLWEIG model (Lindberg et al., 2008), RayMan (Matzarakis et al., 2010), TUF-3D model (Krayenhoff and Voogt, 2007), TUF-IOBES model (Yaghoobian and Kleissl,

---

[1]http://urbanadapt.cz/en

2012, based on TUF-3D), TEB (Masson, 2000) or SUEWS (Järvi et al., 2011). These models treat some physical processes (e.g. radiation, latent heat flux, water balance), while they parameterize the air flow by means of statistical and climatological models or meteorological measurements.

The most complex approach is represented by a group of computational fluid dynamics (CFD) models. The explicit simulation of the turbulent flow is computationally expensive, thus, different techniques have to be adapted to make calculations feasible, usually based on limiting the range of the resolved length and time scales of the turbulent flow. Most of the CFD models applied for urban climatology studies today are models based on the Reynolds-Averaged Navier–Stokes (RANS) equations, e.g. ENVI-met (ENVI-met, 2009), MITRAS (Schlünzen et al., 2003), MIMO (Ehrhard et al., 2000), and MUKLIMO_3 (Sievers, 2012, 2014). In RANS models, the entire turbulence spectrum is parameterized, and thus, only the mean flow is predicted. This allows for using relatively large time steps leading to moderate computational demands, but it implies physical limitations as interactions of turbulent eddies with the urban canopy cannot be explicitly treated. In order to overcome this deficiency, Large-Eddy Simulation (LES) models can be employed. They use a scale separation approach to resolve the bulk of the turbulence spectrum explicitly, while parameterizing only the smallest eddies in a so-called subgrid-scale model. Examples of such models are e.g.: PALM (Maronga et al., 2015), which can incorporate buildings as explicit obstacles; the modelling system OpenFoam[2], which can use both LES and RANS solvers; or DALES (Heus et al., 2010).

However, many of the CFD models do not contain appropriate urban canopy energy balance models with an explicit treatment of radiative fluxes. To overcome this deficiency, stand-alone energy balance models can be coupled to CFD models, recent examples being the SOLENE-microclimat (Musy et al., 2015) or TUF-IOBES, which was coupled to PALM (Yaghoobian et al., 2014). These are usually one-way coupled systems in which the stand-alone model is used for the calculation of incoming/outgoing energy fluxes to/from any surface element, which are then imported into the CFD model. This means that CFD model dynamics are not considered for the calculation of the energy fluxes, making this approach less precise than fully two-way integrated models.

Most of the CFD models are closed-source in-house solutions, complicating their scientific and technical validation. Furthermore, many of them are not designed to work on high-performance computing systems (HPC) with hundreds to thousands of processor cores, limiting their range of applications. Notable exceptions are the models PALM, OpenFoam and DALES, which are available under a free license and can be run on HPC.

Regarding the task at hand, i.e. providing detailed information on the influence of urban surfaces and vegetation on pedestrian-level thermal comfort and air quality, LES models can be considered to be the most appropriate and future-oriented since they can predict the turbulent air flow over a very complex surface with sufficient resolution. However, according to the authors' research at the beginning of the study, there was no open source LES model with an integrated energy balance solver for urban surfaces that would be able to account for the realistic implementation of various processes inside an urban canopy. Our attempts to integrate some of the existing energy models (e.g. TUF-3D) into PALM led to serious technical difficulties due to the different scientific approaches of the particular models, incompatible data structures, difficult parallelization and other issues. The license compatibility was another issue. Therefore, we decided to start from scratch, extending the existing LES

---

[2]http://www.openfoam.org

model PALM with a new fully integrated USM module that explicitly describes energy exchanges in the urban environment. Due to the complexity of this task, the first version of PALM-USM was deliberately limited to the most important processes for modelling summer heat wave episodes in fully urbanized areas. Further improvements and additions to this module are current work in progress and will be realized within the next years (see also Sect. 6).

## 1.2 The LES model PALM

PALM[3] is designed to simulate the turbulent flow in atmospheric and oceanic boundary layers. A highlight of PALM is its outstanding scalability on massively parallel computer architectures (Maronga et al., 2015). The model solves the non-hydrostatic incompressible Navier-Stokes equations in Boussinesq approximation. Subgrid-scale processes that cannot be resolved implicitly based on the numerical grid resolution are parameterized according to the 1.5-order Deardorff closure scheme (Deardorff, 1980) with the modification of Moeng and Wyngaard (1988) and Saiki et al. (2000), assuming that the energy transport by subgrid-scale eddies is proportional to the local gradients of the mean quantities.

Prognostic equations are solved numerically, primarily using an upwind biased fifth-order differencing scheme (Wicker and Skamarock, 2002) and a third-order Runge-Kutta time stepping scheme after Williamson (1980). Discretization in space is achieved using finite differences on a staggered Cartesian Arakawa-C grid (Arakawa and Lamb, 1977).

PALM includes several features such as cloud microphysics, a plant canopy model and an embedded Lagrangian particle model. In connection to the urban application, four other relevant schemes are already implemented: a Cartesian topography scheme, representation of radiative exchange at the surface, large-scale forcing and land-surface interactions with the atmosphere. The Cartesian topography scheme covers solid, impermeable, fixed flow obstacles (e.g. buildings) as well as terrain elevations (mountains, hills), with a constant-flux layer assumed between each surface element and the first grid level adjacent to the respective surface in order to account for friction effects. The representation of radiative exchange at the surface contains options to use either a simple clear-sky radiation parameterization or employing the Rapid Radiation Transfer Model for Global Models (RRTMG, e.g. Clough et al., 2005), which is coupled to PALM and is applied as a single column model for each vertical column in the PALM domain. The large-scale forcing option enables forcing with data e.g. from mesoscale models via additional tendency terms, including an option for nudging of the mean profiles. Finally, the implementation of land-surface interactions with the atmosphere is based on a simplified version of the Tiled European Centre for Medium-Range Weather Forecasts Scheme for Surface Exchanges over Land (TESSEL/HTESSEL, Balsamo et al., 2009) and its derivative implementation on the DALES model (Heus et al., 2010). PALM's land surface submodel (Maronga and Bosveld, 2017), hereafter referred to as PALM-LSM, further extends the surface parameterizations for impervious surfaces on the ground (pavements, roads) by replacing upper soil layers with a pavement layer attributed with a specific heat capacity and heat conductivity.

However, none of the included schemes are suited for treating complex effects of the urban environment driven by the diverse physical properties of different urban surfaces (both horizontal and vertical), heat transfer within building walls, and heat fluxes between the urban surfaces and the atmosphere. Also the description of shortwave and longwave radiation budgets including shading and multi-reflection between surfaces, as well as the absorption of radiation by plant canopies have not been treated

---

[3]https://palm.muk.uni-hannover.de

by PALM so far. Therefore, we developed the USM for PALM that is able to treat these processes using approaches described in the following section.

## 2 Urban surface model

In this section, the first version of the new USM for PALM is described. The USM consists of a solver for the energy balance of all horizontal and vertical urban surface elements, including building walls and roofs, as well as pavements. The energy balance solver predicts the skin layer temperature, and it simultaneously calculates the near-surface turbulent flux of sensible heat and the subsurface conductive heat flux. The latter is calculated by means of a multi-layer model predicting heat diffusion through solid material. Moreover, a multi-reflection radiative transfer model (RTM) for the urban canopy layer was developed, and coupled to the plant canopy model in order to calculate realistic surface radiative fluxes as input for the energy balance solver.

This first version of the USM was designed with the focus on modelling summer heat wave episodes in built-up urban areas. The newly implemented methods hence concentrated on the most relevant processes for such conditions. Limitations of the current version are, e.g.: no treatment of reflective surfaces and windows; only a basic building energy model; simplification of some radiation-related processes (see Sect. 2.2.1 for details); missing plant-canopy evapotranspiration model; surfaces impervious to water. Possible impacts of these limitations are discussed in Sect. 4. Improvements of the USM and related PALM components are subject to ongoing development within the PALM community.

### 2.1 Energy balance solver

The surface energy balance correlates radiative energy fluxes with sensible and latent heat fluxes between the surface skin layer and the atmosphere, as well as with the storage heat fluxes into soil and walls. In this first PALM-USM version, latent heat fluxes were omitted, since the purpose of this version was to simulate heat-wave episodes in fully urbanized areas. This limitation is discussed in Sect. 4.1. The energy budget is expressed in the form:

$$C_0 \frac{dT_0}{dt} = R_n - H - G, \tag{1}$$

where $C_0$ and $T_0$ are heat capacity and temperature of the surface skin layer, respectively, $t$ is the time, $R_n$ is the net radiation, $H$ is the turbulent sensible heat flux near the surface, and $G$ is the heat flux from the surface skin layer into the ground or material (i.e. pavement, walls, roofs). The list of all used symbols, their descriptions and units can be found in supplements in Table S1.

The calculation of the heat transfer $H$ between the surface skin layer and the air is based on the equation:

$$H = h(\theta_1 - \theta_0), \tag{2}$$

where $\theta_0$ is the potential temperature at the surface and $\theta_1$ is the potential temperature of the air layer adjacent to the surface; and $h$ is the so-called heat flux coefficient, which is parameterized for vertical surfaces according to Krayenhoff and Voogt (2007),

while for horizontal surfaces the parameterization of $h$ follows the default PALM-LSM formulation based on Monin-Obukhov similarity theory (Obukhov, 1971). The latter involves the calculation of a local friction velocity, for which stability effects are considered for horizontal surfaces, while stability for vertical surfaces is treated as neutral (i.e., law-of-the-wall scaling is used). The friction velocity is used to calculate both the surface momentum flux for each individual surface element and the coefficient $h$ for horizontal surface elements. The application of MOST for finite-sized surfaces is debatable as the theory is based on the assumption of horizontal homogeneity of the surface and flow, which is violated in urban areas. However, for lack of alternatives, it is the common modelling approach used in all state-of-the-art surface parameterization schemes (e.g. TUF-3D, Krayenhoff and Voogt (2007); SUEWS, Järvi et al. (2011)). The use of MOST in PALM as a boundary condition for buildings has been validated for neutral stratification by Letzel et al. (2008) and Kanda et al. (2013). Moreover, Park and Baik (2013) validated their LES results for non-neutral stratification against wind-tunnel data.

The heat transfer between surface skin layer and subsurface layers follows the general formulation for the heat flux $G$:

$$G = \Lambda(T_0 - T_{\mathrm{matter},1}), \tag{3}$$

where $T_0$ is the temperature of the surface skin layer, $T_{\mathrm{matter},1}$ is the temperature of the outermost layer of the material and $\Lambda$ is an empirical heat conductivity between the skin layer and the first grid level in the material.

The heat transfer within the material layers is calculated via the Fourier law of diffusion. This approach has been generalized for different material types of the pavements, walls, and roofs, each structured into four layers; each layer of each material is described by its own properties (thickness, volumetric thermal capacity and thermal conductivity). The diffusion equation is solved numerically describing the heat transfer from the surface into the inner layers. Boundary conditions of the deepest layer are prescribed in the configuration for particular types of surfaces and are kept constant throughout the simulation. The flux G, calculated in the surface energy balance model, serves as a boundary condition for the outermost material layer.

All non-linear terms in (1) are linearized to avoid the need of an iteration method to solve for the skin temperature (see Maronga and Bosveld, 2017). Equation (1) is then solved by PALM's default Runge-Kutta scheme. The near-surface heat fluxes are evaluated based on the new prognostic skin layer temperature.

## 2.2 Radiative transfer model

### 2.2.1 General concept

The USM receives radiation from the standard PALM solar radiation model at the top boundary of the urban canopy layer. Depending on the chosen radiation module in PALM, the separate direct and diffuse components of the downward shortwave radiation flux may or may not be available. In the latter case, a simple statistical splitting is applied based on Boland et al. (2008). The USM then adds a description of radiation processes in the urban canopy layer where multiple reflections are considered.

Radiation processes are modelled separately for shortwave (SW) and longwave (LW) radiation. Direct and diffuse SW solar radiation along with the relative position of the sun, as well as the LW radiation from the atmosphere is provided by PALM's

solar radiation model. Thermal emission from the ground, walls and roofs is added as a source of longwave radiation. For each time step, radiation is propagated through the 3D geometry of the urban canopy layer for a finite number of reflections, after which all of the radiation is considered as fully absorbed by the surfaces. All reflections are treated as diffuse (Lambertian), and in each reflection, a portion of radiation is absorbed by the respective surface according to its properties (albedo and

5 emissivity). The urban layer may contain arbitrarily located plant canopy (trees and shrubs) described by a 3D structure of leaf area density (LAD), which is treated as semi-opaque for the modelled SW radiation and transforms the absorbed radiation to heat (see Sect. 2.2.4).

Some radiation-related processes have been omitted in this first version, including absorption, emission, and scattering by air within the urban canopy layer, interaction of LW radiation with plant canopy and thermal capacity of plant leaves (plant

canopy is assumed to have the temperature of surrounding air). The effects of these simplifications are discussed in Sect. 4.

### 2.2.2 Calculation of view factors and canopy sink factors

For the calculation of irradiation of each face[4] from diffuse solar radiation, thermal radiation and reflected radiation, mutual visibility between faces of both real surfaces and virtual surfaces (top and lateral domain boundaries) has to be known. It is calculated using a ray tracing algorithm. Since this process is computationally expensive and hard to parallelize (as rays

can travel through the entire domain which is distributed on different processors), both the view factors (SVF) and the plant-canopy sink factors (CSF) are precomputed during the model initialization. These factors can be saved to file and used for other simulations with the same surface geometry, or for the calculation of the mean radiant temperature (MRT) in the postprocessing.

For any two faces $A$ and $B$ with mutual visibility, the *view factor* $F_{A \to B}$ represents the fraction between that part of radiant flux from face $A$ that strikes face $B$ and the total radiant flux leaving face $A$. For infinitesimally small areas of $A$ and $B$, a

20 *differential view factor* $F^d_{A \to B}$ can be written as:

$$F^d_{A \to B} = \frac{dF_{A \to B}}{d\mathcal{A}(B)} = \frac{\cos\theta_A \cos\theta_B}{\pi s^2}, \tag{4}$$

where $\mathcal{A}(B)$ is the surface area of face $B$, $\theta_A$ and $\theta_B$ are the angles between the respective face normals and the connecting ray, and $s$ is the separation distance (ray length) (Howell et al., 2010), see Fig. 1a. Under the assumption that $s$ is much larger than the grid resolution, differential view factors are precomputed for all mutually visible face centres and used in place of

25 view factors divided by target area. At the end, the differential view factors are normalized such that the sum of all normalized differential view factors with the same target face ($B$) multiplied by source area equals 1:

$$\widehat{F}^d_{A \to B} = \frac{F^d_{A \to B}}{\sum_{A'} F^d_{A' \to B} \mathcal{A}(A')}. \tag{5}$$

If the view factors were known exactly, the sum $\sum_{A'} \frac{F_{A' \to B} \mathcal{A}(A')}{\mathcal{A}(B)}$ would equal 1 (from the reciprocity rule $\mathcal{A}(A)F_{A \to B} = \mathcal{A}(B)F_{B \to A}$). Therefore, the normalization guarantees that, in total, no radiation is lost or created by simplification due to

30 discretization. Since the part of face $B$'s irradiance that comes from face $A$ is computed as

$$J_{e,A \to B} = E_{e,A} \mathcal{A}(A)\widehat{F}^d_{A \to B}, \tag{6}$$

---

[4]A face is a unit of surface according to discretization by grid; it is a boundary between a grid box with terrain/building and an adjacent air-filled grid box.

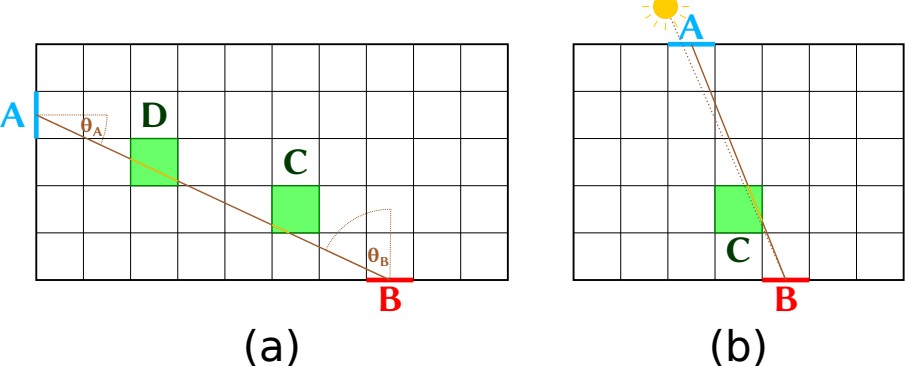

**Figure 1.** (a) View factor calculation (2-D simplification); (b) Direct solar irradiation

where $E_{e,A}$ is the radiosity of face $A$, we specifically precompute and store the value of

$$SVF_{A\to B} = \mathcal{A}(A)\widehat{F}^d_{A\to B} \tag{7}$$

which we call the *irradiance factor*. In case the ray tracing algorithm encounters an obstacle (i.e. wall or roof), the view-factor entry is not stored, indicating absence of mutual visibility between the two respective faces.

5  The equations above describe radiative fluxes before accounting for plant canopy. For every ray that crosses a grid box containing plant canopy (i.e. a partially opaque box), a dimensionless *ray canopy sink factor* (RCSF) represents the radiative flux absorbed within the respective grid box normalized by the total radiative flux carried by the ray at its origin. For a ray $A \to B$ and a grid box C, the RCSF is calculated as

$$RCSF_{C,A\to B} = \left(1 - \sum_D RCSF_{D,A\to B}\right)\left(1 - e^{-\alpha a_C s_C}\right), \tag{8}$$

10 where $a_C$ is the leaf area density of grid box $C$, $s_C$ is the length of the ray's intersection with box $C$ and $\alpha$ is the extinction coefficient. The sum in the first term represents cumulative absorption by all plant-canopy-containing grid boxes $D$ that have already been encountered on the ray's path before reaching grid box $C$ (Fig. 1a).

  After the entire ray is traced, the total transmittance $T$ of the ray $A \to B$ passing through plant canopy grid boxes $C$

$$T_{A\to B} = 1 - \sum_C RCSF_{C,A\to B} \tag{9}$$

15 is stored along with $SVF_{A\to B}$. Later in the modelling, when the radiant flux transmitted through $SVF_{A\to B}$ is calculated, it is multiplied by $T_{A\to B}$ to account for the absorbed flux.

  The actual radiant flux $\Phi_e$ received by the grid box $C$ from the ray $A \to B$ is equal to

$$\Phi_{e,C,A\to B} = E_{e,A} \cdot SVF_{A\to B} \cdot \mathcal{A}(B) \cdot RCSF_{C,A\to B}. \tag{10}$$

The radiosity $E_{e,A}$ of the source face is the only time-dependent variable in this equation. Therefore, the rest of this product can be precomputed during initialization, and summed up per source face in the form of a *canopy sink factor* (CSF):

$$CSF_{C,A} = \sum_B SVF_{A \to B} \cdot \mathcal{A}(B) \cdot RCSF_{C,A \to B}. \tag{11}$$

CSF represents the ratio between the radiant flux absorbed within plant canopy box $C$ originating from face $A$ and the radiosity of face $A$.

### 2.2.3 Calculation of per-face irradiation

At each time step, the total incoming and outgoing radiative fluxes of each face are computed iteratively, starting from the first pass of radiation from sources to immediate targets, followed by consecutive reflections.

In the first pass, the virtual surfaces (sky and domain boundaries) are used as sources of radiation by representing components of diffuse shortwave solar radiation and longwave radiation from the sky. At this point, the real surfaces (wall facades, roofs, ground) are set to emit longwave radiation according to their surface temperature and emissivity. The precomputed view factors are then used to cast the shortwave and longwave radiation from source to target faces.

Solar visibility has to be calculated for the quantification of the direct part of shortwave solar radiation. The solar angle is discretized for this purpose so that the solar ray always originates from the center of the virtual face at the top of the urban layer or at lateral domain boundaries (see the real location of sun vs. discretized location (center of face A) in Fig. 1b). We have decided not to do the computationally expensive ray tracing after the precomputation phase is over, moreover, the total transmittance stored alongside the precomputed view factor (see Eq. (9)) is readily available. If there is no such view factor entry, it means that the discretized ray path is blocked by a wall or roof and the target face receives no direct solar irradiation. For the purpose of calculating the actual amount of direct solar irradiation, an exact solar angle is used, not the discretized one.

After the aforementioned first pass of radiation from source to target surfaces has been computed, reflection is applied iteratively. At each iteration, a fraction of each surface's irradiation from the previous iteration is reflected and the remainder is considered absorbed. The reflected fraction is determined by the surface's albedo for shortwave radiation and by the surface's emissivity $\varepsilon$ for longwave radiation, where the longwave reflectivity results from $(1 - \varepsilon)$ according to Kirchhoff's law. The reflected part is then again distributed onto visible faces using the precomputed view factors. After the last iteration, all residual irradiation is considered as absorbed. The number of iterations is configurable, and the amount of residual absorbed radiation can be displayed in the model output. In our experience, three to five iterations lead to negligible residue.

### 2.2.4 Absorption of radiation in the plant canopy

The fraction of SW radiative flux absorbed by the plant canopy is calculated for the first pass as well as for all the successive reflection steps (these are described in Sect. 2.2.3).

For diffuse and reflected shortwave radiation, the amount of radiative flux absorbed by each grid box with plant canopy is determined using the precomputed CSF and radiosity of the source face (i.e. reflected radiosity for a real surface or diffuse solar irradiance for a virtual surface, see Eq. (10)).

For the direct solar irradiance, the nearest precomputed ray path from the urban-layer bounding box (represented by virtual face $A$ in Fig. 1b) to the respective plant canopy grid box $C$ is selected similarly as for the direct surface irradiation described in Sect. 2.2.3. In case the grid box $C$ is fully shaded, no ray path is available. Otherwise, the transmittance of the path is known. The absorbed direct solar flux for the grid box $C$ is equal to

$$\Phi_{e,C} = E_{e,dir} \cdot T_{A \to C} \cdot \frac{\iint_b (1 - e^{-\alpha a_C s_b})\, \mathrm{d}b}{\mathcal{A}'_C}, \tag{12}$$

where $E_{e,dir}$ is the direct solar irradiance and $\mathcal{A}'_C$ is the cross-sectional area of $C$ viewed from the direction of the solar radiation. The fraction in Eq. (12) represents the absorbed proportion of radiative flux, averaged over each ray $b$ that intersects the grid box $C$ and is parallel to the direction of the solar radiation; $s_b$ is the length of the intersection. Since all grid boxes have the same dimensions, this fraction is precomputed based on the solar direction vector at the beginning of each time step using discrete approximation.

Once the total absorbed radiative flux is known, it is stored as plant canopy heat rate for the respective grid box. Since the plant canopy is considered to have zero thermal capacity, all of the heating power is applied immediately to the grid box's air volume.

## 2.3 Anthropogenic heat

The prescribed anthropogenic heat is assigned to the appropriate layer of the air, where it increases the potential-temperature tendencies at each time step. This process takes place after the surface energy balance is solved. The heat is calculated from daily total heat released into any particular grid box, and from the daily profile of the release specified for every layer to which anthropogenic heat is released.

## 2.4 USM module integration into PALM

The USM was fully integrated into PALM, following its modular concept, as an optional module, which directly utilizes the model values of wind flow, radiation, temperature, energy fluxes and other required values. The USM returns the predicted surface heat fluxes back to the PALM core, where they are used in the corresponding prognostic equations. It also adjusts the prognostic tendencies of air according to released anthropogenic heat.

Descriptions of real and virtual surfaces and their properties are stored in one dimensional arrays indexed to the 3D model domain. The crucial challenge of this part of the design is to ensure an efficient parallelization of the code, including an efficient handling and access of data stored in the memory during the simulation. The values are stored locally in particular processes of the Message Passing Interface (MPI[5]), corresponding to the parallelization of the PALM core. Necessary access to values stored in other processes is enabled by means of MPI routines, including interfaces for one-sided MPI communication.

The configuration of the USM module is compatible with other PALM modules. Variables for instantaneous and time-averaged outputs of the USM are integrated into PALM's standard 3D NetCDF output files, and they are configured in the

---

[5]http://mpi-forum.org

same way as the rest of the model output variables. The configuration options as well as the structure of input and output files are described in the supplements to this article. The supplement also contains the list and description of needed surface and material parameters of urban surfaces, plant-canopy data and anthropogenic-heat data. The model PALM with its USM module is hereafter referred to as PALM-USM, which is freely available under the GNU General Public License (see Code availability section).

## 3 Evaluation and sensitivity tests of USM

In order to evaluate how well the USM represents urban surfaces' temperatures (of e.g. walls, roofs, streets), a summer heatwave observation campaign in an urban quarter of Prague, Czech Republic, was carried out (see Sect. 3.1). By means of PALM-USM, urban-quarter characteristics and the campaign's meteorological conditions were simulated (see Sect. 3.2), and model results evaluated against the observations (see Sect. 3.3).

### 3.1 Observation campaign

The campaign was carried out at the crossroads of Dělnická street and Komunardů street in Prague, Czech Republic (50.10324N, 14.44997E, terrain elevation 180 m above sea level (ASL)). This location was selected in coordination with the Prague Institute of Planning and Development as a case study area for urban heat island adaptation and mitigation strategies. This particular area represents a typical residential area in a topographically flat part of the city of Prague with a combination of old and new buildings and a variety of other urban components (such as yards or parking spaces). The streets are oriented in a north-south (Komunardů) and west-east (Dělnická) direction, roughly 20 and 16 m wide, respectively. The building heights alongside the streets range approximately from 10 to 25 m. The area does not contain much green vegetation and the majority of the trees is located in the yards. The neighbourhood in the extent of approximately 1 km$^2$ has similar characteristics as the study area (see aerial photo in Fig. 2).

Measurements were conducted from 2 July 2015, 14:00 UTC to 3 July 2015, 17:00 UTC. The timing of the measurement campaign was chosen to cover a typical summer heatwave episode.

### 3.1.1 Measurements

Wall and ground surface temperatures were measured by an infrared camera FLIR SC660 (FLIR, 2008). The thermal sensor of the camera has a field of view of 24° by 18° and a spatial resolution (given as instantaneous field of view) of 0.65 mrad. The spectral range of the camera is 7.5 to 13 μm, and the declared thermal sensitivity at 30 °C is 45 mK. The measurement accuracy for an object with a temperature between +5 °C and +120 °C, and given an ambient air temperature between +9 °C and +35 °C, is ±1 °C or ±1 % of the reading. The camera offers a built-in emissivity-correction option, which was not used for this study. Apart from the infrared pictures, the camera allows to take pictures in visible spectrum simultaneously.

Observation locations are shown in Fig. 3, eight of them (No. 1–7 and 9) representing temperature measurements of the walls on the opposite side of the street, and one of them (No. 8) representing the ground-temperature measurement on the

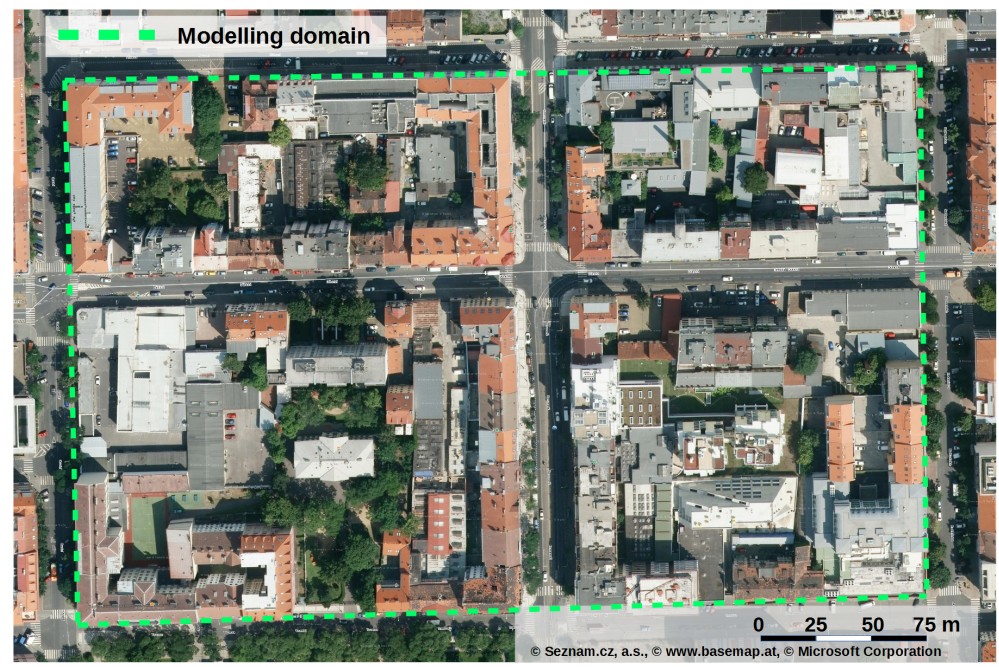

**Figure 2.** Aerial photo of the studied area.

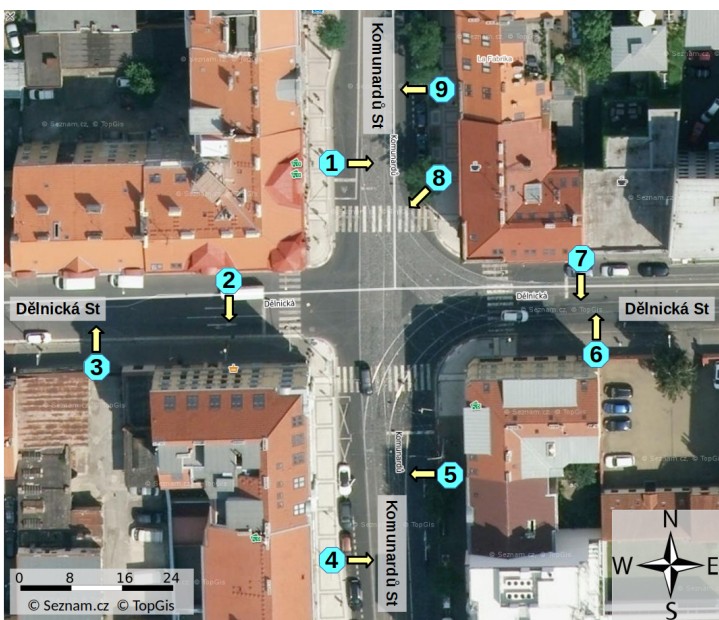

**Figure 3.** Observation locations. Arrows depict the orientation of the camera view. Url of the map: https://mapy.cz/s/12Qd8.

road. The camera was always placed on a tripod at about 1.6 m above ground, with its orientation approximately perpendicular to the opposite walls, at a distance of 14–18 m depending on the street width. For scans of the ground, the picture's centre was viewed under the angle of about 15° (closest point was approx. 2 m away and the most distant point 30 m, resulting in view angles of 38° and 3°, respectively). Pictures in both the infrared and visible spectrum were taken simultaneously, starting at observation location 1 every full hour and continuing through observation locations 2, 3, etc. This provided a series of 28 temperature snapshots per location with approximately 1 h time step. The exact recording time of each picture was used for further processing and evaluation of the model.

The pictures were further postprocessed. First, the infrared pictures were converted into a common temperature scale +10 °C to +60 °C. Second, the pictures were transformed to overlap each other in order to correct for slight changes in camera position between the measurements, as the camera was carried from one location to another. Third, several evaluation points were selected for each view to cover various surface types in order to evaluate the model performance under different surface parameter settings (different surface materials and colours) and under different situations (fully irradiated or shaded areas). Namely, selected surface materials comprised old and new plastered brick house walls as well as modern insulated facades for vertical surfaces, and pavement or asphalt for ground observation location. With regard to colours, the evaluation points were placed on both dark and light surfaces, with special interest in places where light and dark materials are located side-by-side, thus allowing to inspect different albedo settings under roughly the same irradiance conditions. Some points were placed on wall areas, which are temporarily (in the diurnal cycle) shaded by trees or buildings, in order to test how the shading works in the model.

Apart from infrared camera scanning, the air temperature was measured once an hour at observation location 1 at the edge of the pavement, about 2 m apart from the wall and 2 m above ground, not in direct sunlight. A digital thermometer with external NTC-type thermistor measuring probe (resolution of 0.1 K and declared accuracy of 1 K) was used. This on-site measurement did not meet requirements for the standard meteorological measurement, therefore we refer to it as indicative measurement later in the text.

Further meteorological data was acquired from the official weather monitoring network, including stations Prague, Klementinum; Prague, Karlov; Prague, Kbely and Prague, Libuš. Station Prague, Klementinum (50.08636N, 14.41634E, terrain elevation 190 m ASL, 3 km away from the crossroads of interest) was used as supplementation to the on-site indicative measurement. The temperature at this station is measured on the north facing wall, 10 m above the courtyard of the historical building complex, and it can be used as another reference for the air temperature inside the urban canopy. Station Prague, Karlov (50.06916N, 14.42778E, 232 m ASL, 4.3 km away) can be considered as representative for the city core of Prague as it is located in the centre of the city. Station Prague, Kbely (50.12333N, 14.53806E, 285 m ASL, 6.7 km away) is located at the border of the city and serves as a reference for regional background suburban meteorological conditions. Station Prague, Libuš (50.00778N, 14.44694E, 302 m ASL, 10 km away) is located at the city suburb and it is the only station with sounding measurements in the area. Radiosondes are released three times a day (00:00, 06:00, 12:00 UTC).

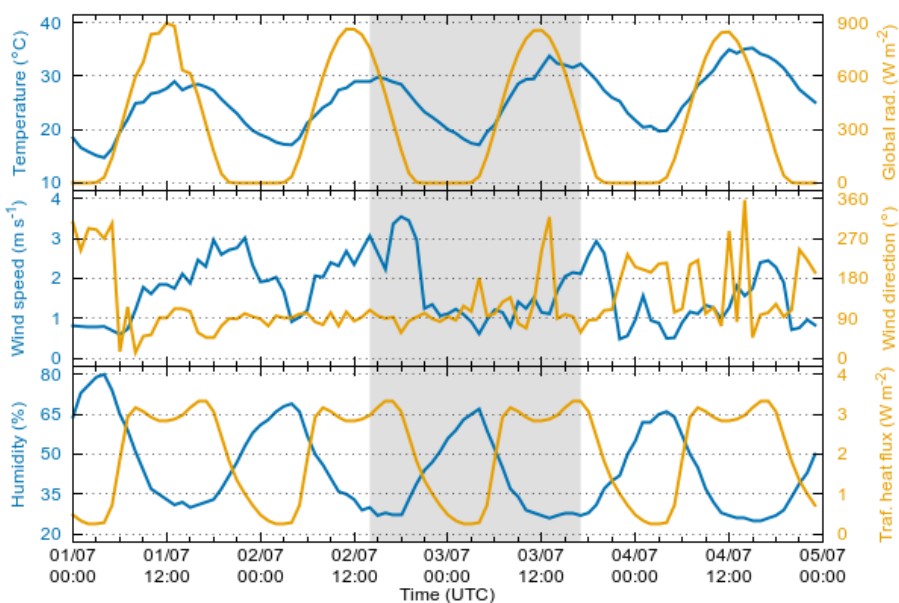

**Figure 4.** Meteorological conditions from the station Prague, Karlov, and spatially averaged traffic heat flux from 1 July to 4 July. The shaded area marks the time of the observation campaign.

### 3.1.2 Weather conditions

The weather during the campaign was influenced by a high-pressure system centred above the Baltic Sea. The meteorological conditions at Prague, Karlov station are shown in Fig. 4. Winds above rooftop were weak, mostly below 2.5 m s$^{-1}$ and often as low as 1 m s$^{-1}$ from easterly directions. Maximum measured wind speeds of 3–4 m s$^{-1}$ were observed in the afternoons at the beginning and at the end of the campaign. According to the atmospheric sounding, a low-level jet from south and south-east was observed during the night, with a maximum wind speed of 10 m s$^{-1}$ at 640 m ASL (950 hPa) (not shown). The temperature exceeded 30 °C in the afternoons and dropped to 20 °C at night. The sky was mostly clear with some clouds during the daytime on 1 July and high-altitude cirrus forming in the morning and afternoon on 3 July. Highest values of relative humidity occurred at night (65 %), dropping to 30 % during the day. The time of the sunset was 19:15 UTC on 2 July 2015, and the time of sunrise and solar noon on 3 July was 02:58 UTC and 11:06 UTC.

### 3.2 Model setup and input data for USM

To assess the validity of the model formulation and its performance in real conditions, the model was setup to simulate the measured summer episode described in Sect. 3.1. The total simulation time span was 48 hours, starting on 2 July 2015, 00:00 UTC.

### 3.2.1 Model domain

The horizontal size of the model domain was 376 m × 226 m (see Fig. 2) at a resolution of 2.08 m × 2.08 m. The vertical grid spacing was 2.08 m within the first 50 m, and above this level, a vertical stretching factor of 1.08 between two adjacent levels was used with a maximum grid spacing of 20 m. The total domain height resulted to about 3.5 km. The relatively small horizontal model domain was chosen due to available data of surface parameters and to keep computational demands feasible. This poses some limitation to the turbulence development during the daytime, where the largest eddies usually scale with the height of the boundary layer. These eddies could not be captured well with this configuration. The effects of the limited horizontal size of the domain on model results will be discussed in Sect. 4.

### 3.2.2 Boundary and initial conditions

Lateral domain boundaries were cyclic, which can be envisioned as an infinite repetition of the simulated urban quarter. This is a reasonable approximation, since the surrounding area has similar characteristics as the model domain, thus, the character of the flow can be considered similar. The bottom boundary was driven by the heat fluxes as calculated by the energy balance solver (see Sect. 2.1). At the top of the domain, Neumann boundary conditions were applied for potential temperature and relative humidity, while a Dirichlet boundary was set for the horizontal wind. A weak Rayleigh damping with a factor of 0.001 was applied to levels above 3000 m. The indoor temperature was fixed at 22 °C during the entire simulation.

The initial vertical profile of potential temperature of air was derived from the sounding measurement at the outskirt area Prague, Libuš station from 2 July 2015, 00:00 UTC (see Fig. 5). At midnight, a stable layer had developed near the surface, extending to a height of about 300 m. Above, a residual layer with slightly stable stratification ranged up to the inversion at around 1900 m. The capping inversion had a strength of about 5 K, with the stable free atmosphere aloft. The temperature of walls, grounds and roofs was initialized from a 24-hour spin-up simulation.

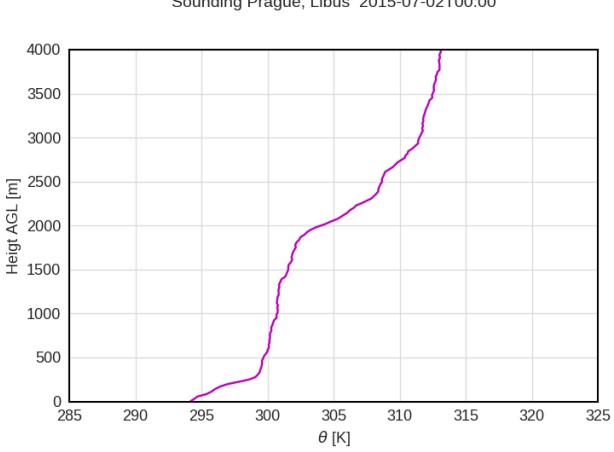

Sounding Prague, Libuš 2015-07-02T00:00

**Figure 5.** Initial vertical profile of potential temperature ($\theta$) as used for initialization of PALM.

### 3.2.3 Large-scale forcing

To account for the processes occurring on larger scales than the modelling domain, but still affecting the processes inside the domain, the large-scale forcing option of PALM was used. The effect of large-scale conditions is included via geostrophic wind and large-scale advection tendencies for temperature and humidity. The forcing quantities can depend on both height and time, while being horizontally homogeneous. Nudging of PALM quantities towards the large-scale conditions was enabled for the free atmosphere layers with a relaxation time of 7 hours. Inside the boundary layer nudging was disabled. Large-scale forcing and nudging data were generated based on a run with the meso-scale numerical weather prediction model WRF (version 3.8.1, Skamarock et al. (2008)). The WRF simulation domain covered a large part of Europe ($-1.7$–$34.7°$ longitude, $41.4$–$56.7°$ latitude, $9$ km horizontal resolution, 49 vertical levels). Standard physics parameterizations were used including the RRTMG radiation scheme, Yonsei University PBL scheme (Hong et al., 2006), Monin-Obukhov similarity surface layer and Noah land surface model (Tewari et al., 2004). The urban parameterization was not enabled, in order to avoid double counting of the urban canopy effect which is treated by PALM-USM. The configuration of the WRF model corresponds to the prediction system routinely operated by the Institute of Computer Science of The Czech Academy of Sciences[6].

The output of the WRF model was compared to measurements from the four Prague stations (see Sect. 3.1.1). The overall agreement between the simulated values and the observations is reasonable and corresponds with long term evaluations done earlier. For the period of 1–5 July (see Fig. S1), WRF shows a cold bias. The largest bias occurs in the urban Prague, Klementinum station (city centre), which is as expected given the urban parameterization not being enabled in the WRF model. On the other hand, the comparison with Prague, Kbely station (closest background station to the area of interest) shows only a small bias (see also timeseries in Fig. S2). Also the comparison with vertical profiles of temperature from Prague, Libuš station shows good agreement (see Fig. S3). Despite the slight cold bias of the WRF simulation, we take the WRF-derived values as the best inputs available.

### 3.2.4 Surface and material parameters

Solving the USM energy balance equations requires a number of surface (albedo, emissivity, roughness length, thermal conductivity, and capacity of the skin layer) and volume (thermal capacity and volumetric thermal conductivity) material input parameters. When going to such a high resolution as in our test case ($\sim 2$ m), the urban surfaces and wall materials become very heterogeneous. Any bulk parameter setting would therefore be inadequate. Instead, we opted for a detailed setting of these parameters wherever possible. To obtain these data, a supplemental on-site data collection campaign was carried out and a detailed database of geospatial data was created. This includes information on wall, ground and roof materials and colours which was used to estimate surface and material properties. Each surface is described by material category and albedo. Categories are assigned to parameters estimated based on surface and storage material composition and thickness. The parameters of all subsurface layers of the respective material were set to the same value. The parameters $C_0$ and $\Lambda$ (see Eq. 1 and 2) of the skin layer are inferred from the properties of the material near the surface, which may differ from the rest of the volume.

---

[6]http://medard-online.cz/

Parameters associated to particular categories are given in supplements Table S2. A tree is described by its position, diameter and vertically stratified leaf area density. Building heights were available from the Prague 3D model, maintained by Prague Institute of Planning and Development[7]. All descriptions of surfaces and materials and their properties were collected in GIS formats and then preprocessed into the USM input files corresponding to the particular domain setup.

### 3.2.5 Anthropogenic heat

Anthropogenic heat sources for our particular case are dominated by heat from fuel combustion in cars (see also discussion in Sect. 4). Based on Sailor and Lu (2004), we assume the average heat release to be 3975 J per vehicle per meter of travel. Traffic intensities and hourly traffic factors are based on the annual traffic census data. The traffic intensities vary for different arms of the crossroads and traffic directions. The total count of vehicles passing through the crossroads is 12000 vehicles per day, and the intensity of the busiest road (west arm of west-east street) is 6000 vehicles per day. The heat produced by the cars along their trajectories is released into the first model layer and spatially distributed into the model grid cells that correspond to the traffic lanes. Temporal distribution is done using prescribed hourly factors. The time factors are the same for all traffic lanes. Values of anthropogenic heat are 42 W m$^{-2}$ on average (spatially and temporally), while the maximum value is 142 W m$^{-2}$ (busiest road arm, peak hour). Those values refer to heat fluxes directly above the traffic lanes. The mean daily traffic heat flux averaged over the entire domain is 2 W m$^{-2}$. The daily course of the traffic heat release is plotted on Fig. 4. It has been shown before that for this particular case (with strong solar irradiance, high temperatures and only moderate traffic) the inclusion of anthropogenic heat from transportation does not result in a noticeable change in temperatures and heat fluxes (Juruš et al., 2016).

## 3.3 Evaluation of PALM-USM

First we compare the air temperatures from PALM-USM to the measurements taken during the observation campaign. Figure 6 shows the air temperature course calculated by the PALM-USM at observation location 1 at 2 m above ground. This temperature is compared to the indicative measurement taken at the same place and also to the automatic weather stations Prague, Klementinum, and Prague, Kbely. The indicative measurement together with station Prague, Klementinum represent the conditions inside the urban canopy, and as such, the results of PALM-USM should correspond to those values. Prague, Kbely station is plotted as a representative of the outskirts of the city. The UHI effect is clearly visible, especially at night, when the temperature outside of Prague drops down to 15 °C while on the street level, it drops to 20 °C only. This effect is less pronounced during the day when the temperature difference is only 2–3 °C. This reflects the known fact that the UHI is basically a nighttime effect (Oke, 1982). The street-level air temperature as simulated by PALM-USM is in agreement with both measurements during the daytime of 2 July, but starting from 21:00 UTC, the decrease of the modelled temperature weakens, gradually leading to overestimations of up to 2 °C in the morning of 3 July.

The vertical structure of the potential temperature from PALM-USM is shown together with radiosonde observations from station Prague, Libuš in Fig. 7. As this is a suburban background station, (10 km away) the profiles are not truly comparable,

---

[7]http://www.geoportalpraha.cz

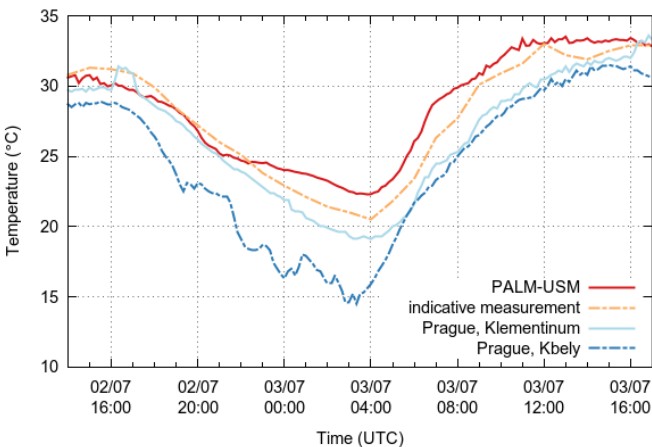

**Figure 6.** Air temperatures obtained from PALM-USM for location 1 in comparison to measured temperatures.

especially near the surface, where effects of the UHI are expected in the PALM-USM data. The Libuš profiles are considered here mainly as a representation of the general meteorological situation in the area of interest.

From Fig. 7 it is well visible that PALM-USM was able to reproduce the diurnal temperature cycle of the boundary layer reasonably well. During daytime, a convective boundary layer (CBL) develops that reaches depths of about 2 km, which is
somewhat higher than the observed boundary layer, particularly on 2 July (12:00 UTC). This can be attributed to the higher amount of heat released by the surfaces of densely built-up urban areas, as compared to the surfaces of suburban regions where the radiosonde was released. Moreover, it is visible that the potential temperature profile produced by PALM-USM displays an unstable stratification throughout the CBL on both days, while the observations show the expected nearly-neutral profile. We will later see that this is an effect of the limited horizontal model domain that inhibits the free development of the largest
eddies and thus is limiting the vertical turbulent mixing of warm air from the surface and relatively cold air from above. The result is an unstable layer with somewhat too high near-surface temperatures.

During nighttime, a stable boundary layer developed in both, LES and observations (due to nocturnal radiative cooling). As expected, this cooling is more rigorous in the (suburban) measurements so that the stable layer was able to extend to heights of 500 m, whereas PALM-USM predicts a stable layer of not more than 100 m vertical extent (see 00:00 UTC on 3 July and 4
July). This result is in agreement with what was already shown in Fig. 6 and is a known feature of the UHI (Oke, 1982).

Figures 8a–c show the temporal development of the turbulence, which is here represented by the variance of vertical velocity. The diurnal turbulence cycle is very well visible, with maximum intensities of 1.4 $m^2 s^{-2}$ around noon, located in the well-mixed part of the boundary layer (Fig. 8c). Ideally, it would show a clear maximum in the middle of the boundary layer, but two processes avoid this. First, the urban canopy arranges the release of heat at different heights above the ground surface, and
second, the limited horizontal model domain does not allow for a free development of turbulence. Figure 8a further shows that the turbulence immediately starts to decay after sunset which is accompanied by the development of a stable boundary layer near the surface (not shown here). During nighttime, the turbulence further decays and the maximum values of the variance

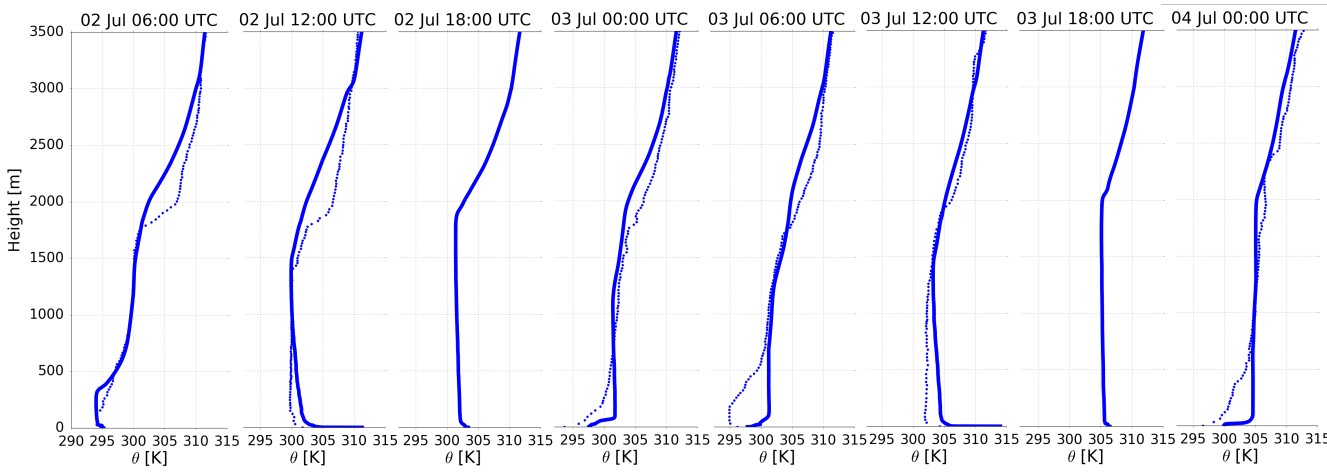

**Figure 7.** The vertical profiles of potential temperature modelled by PALM (solid line) and supplemented by radiosonde observations from station Prague, Libuš (except hour 18; dotted line). Displayed are profiles from 2 July 06:00 UTC to 4 July 00:00 UTC with 6-hour time step

.

reduce to 0.3 $\mathrm{m^2\,s^{-2}}$. Due to the continuously heating urban surface layer, however, turbulence is kept alive until the next morning (see also Fig. 8a).

Next, in order to assess how well the model is able to model the energy transfer between material and atmosphere, we compare modelled values to on-site measurements of surface temperatures captured by the infrared camera. Here we present
results from five selected locations, chosen to cover wall orientations to all cardinal directions and the ground: location 3 (south facing wall) in Fig. 9, location 4 (west facing wall) in Fig. 10, location 7 (north facing wall) in Fig. 11, location 9 (east facing wall) in Fig. 12 and location 8 (ground) in Fig. 13. Corresponding surface and material parameters for all evaluation points can be found in Tables S2 and S3 in supplements. Results for all nine locations are also displayed in supplements (Figs. S4–S12). In general, PALM-USM captures the observed daily temperature patterns very well. The temperature values during the daytime
are captured reasonably well, while the model slightly overestimates nighttime temperatures.

Figure 9 shows a south facing wall in the west arm of the west-east street measured from location 3. We evaluated the model performance in four points. All points are assigned the same material category (plastered brick wall, see Tables S2 and S3). Points 1, 2 and 3 are on a surface with the same colour, which is represented by an albedo of 0.2 in the model. Point 4 is placed on a surface of lighter colour (albedo of 0.7). The lighter surface colour in point 4 results in a significantly lower measured peak
temperature of 6–9 °C less than in other points. The model correctly captures the lower temperature in the point 4, although the modelled maximum in point 4 is a bit higher than the measured maximum. The effect of different albedos can be seen in Fig. 13 for points 2, 3 and 4, too. The observation that the model overestimates values at some evaluation points located in the lowest parts of the buildings can also be made at other observation locations (see location 4, Fig. 10, point 1 or location 5, Fig. S8, point 1).

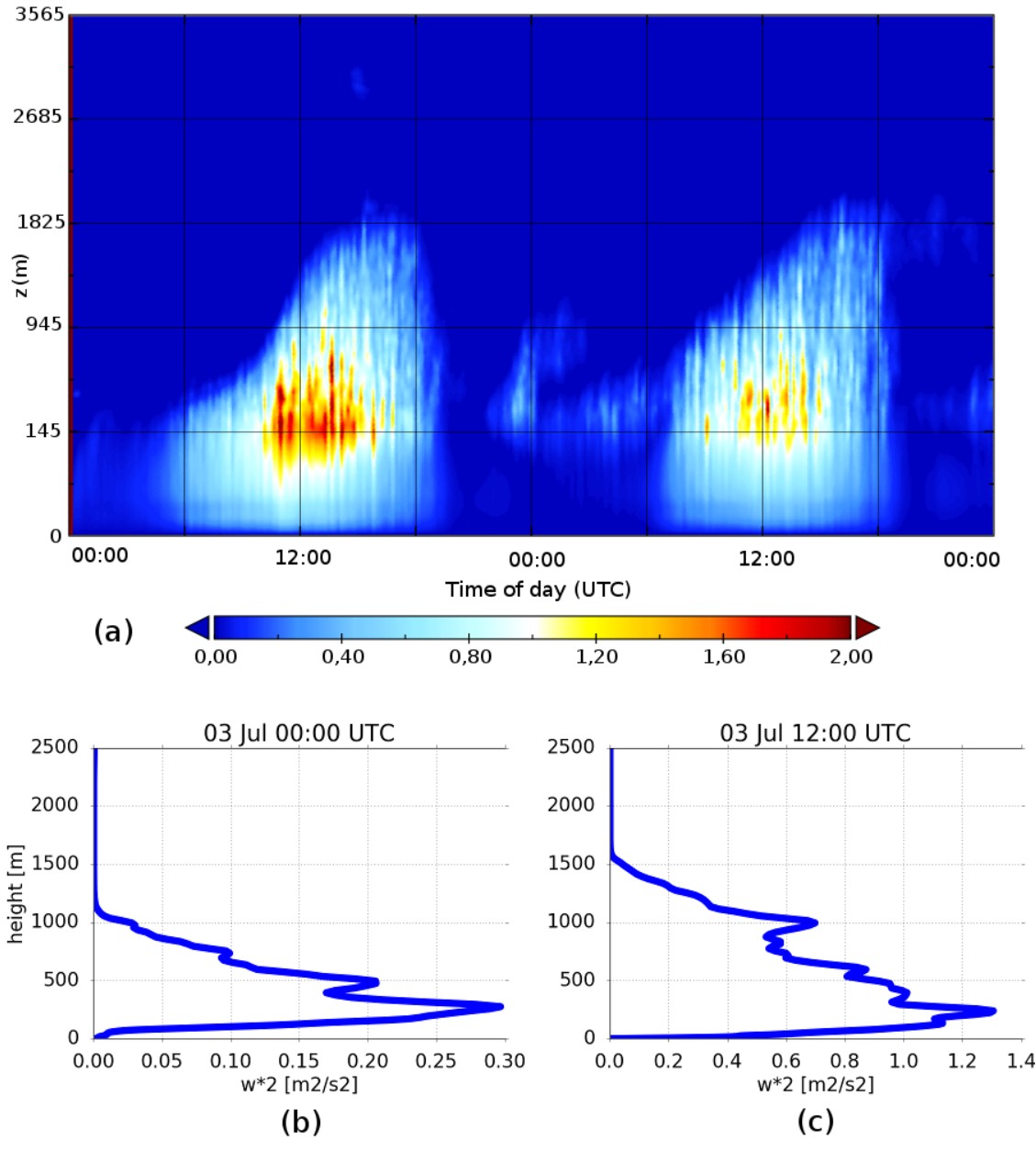

**Figure 8.** (a) Time-height cross section of the variance of the vertical velocity component (top), and vertical profiles of the same quantity at two selected times: (b) 3 July 00:00 UTC and (c) 3 July 12:00 UTC (bottom).

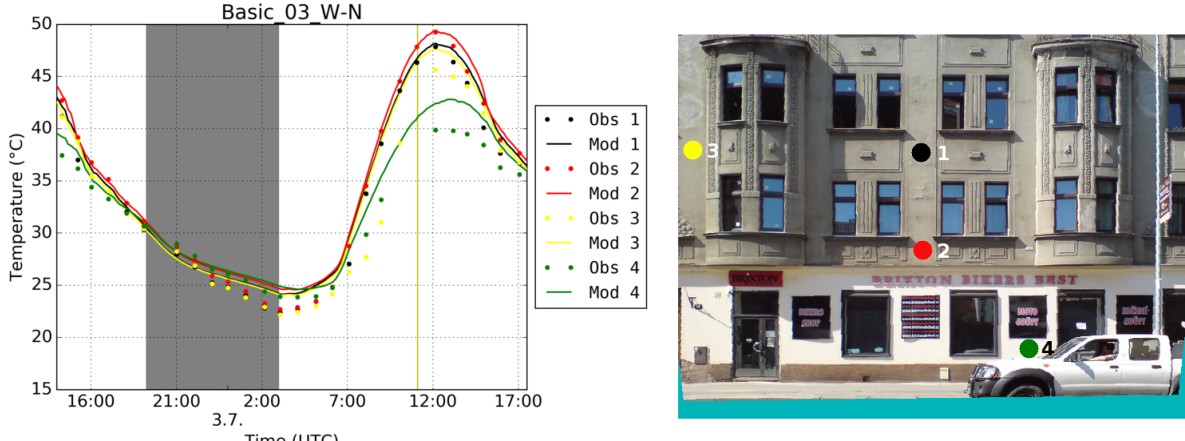

**Figure 9.** Comparison of modelled and observed surface temperatures from the observation location 3 (50.10354N, 14.45006E) — view of south facing wall. The graph shows comparisons for selected evaluation points for the period of the observation campaign from 2 July 2015, 14:00 UTC to 3 July 2015, 17:00 UTC. The solid line represents modelled values while the dots show the observed values. The shaded area depicts nighttime and the yellow vertical line depicts solar noon.

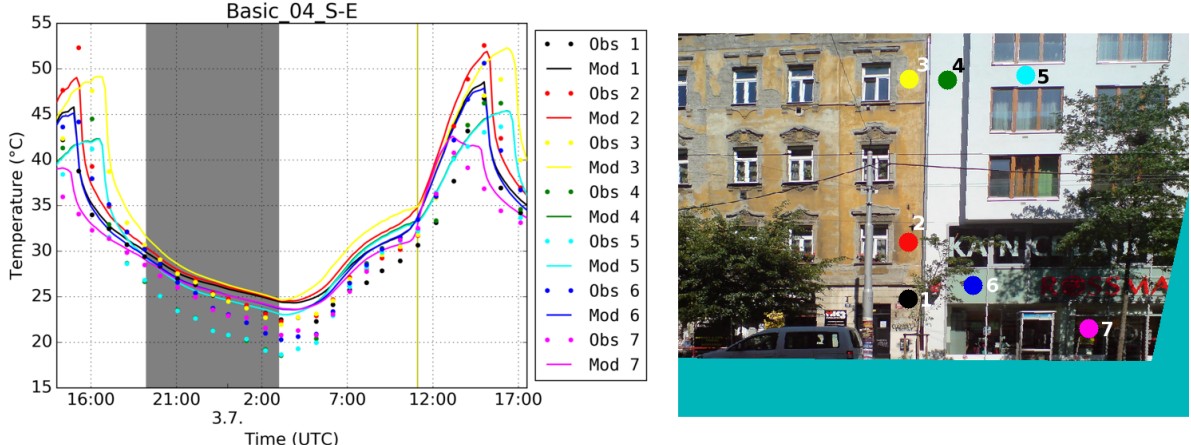

**Figure 10.** As Fig. 9 for location 4 (50.10288N, 14.44985E) — view of west facing wall.

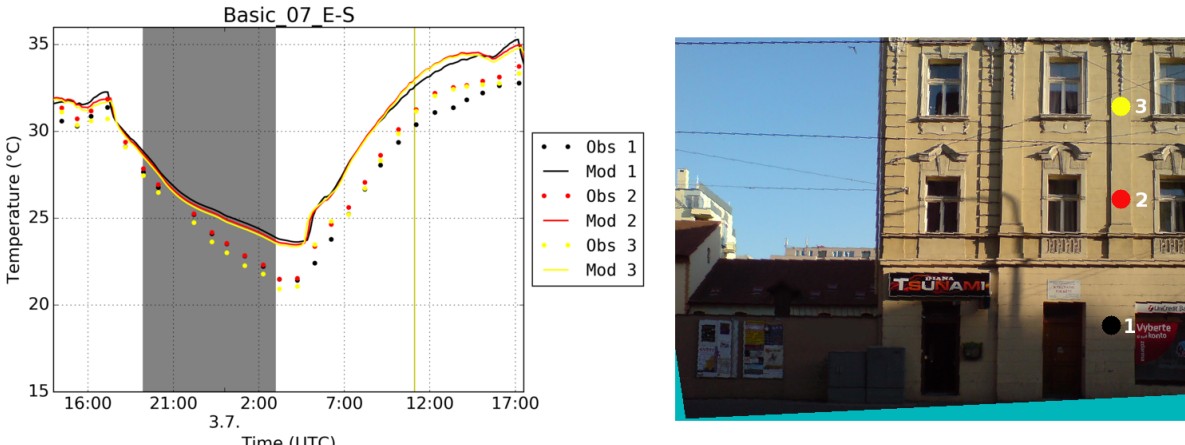

**Figure 11.** As Fig. 9 for location 7 (50.10329N, 14.45040E) — view of north facing wall.

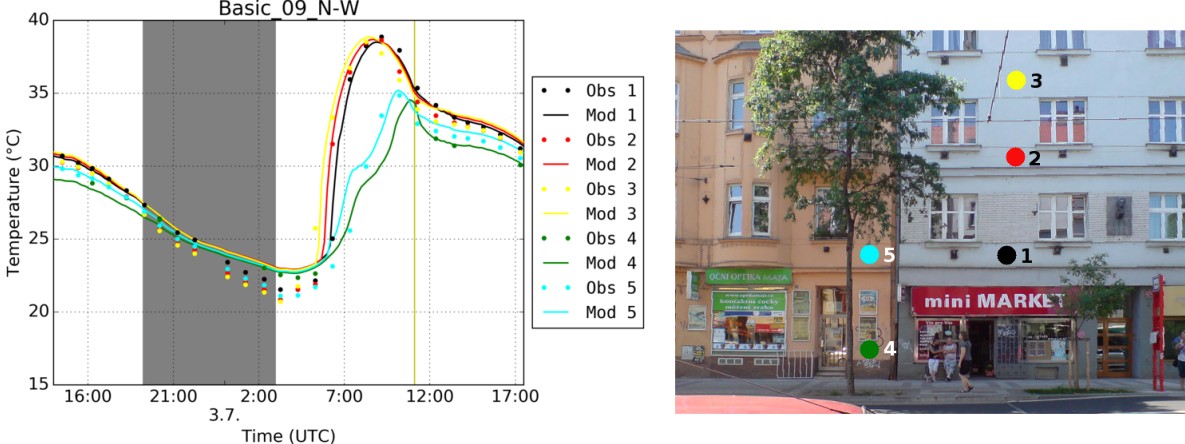

**Figure 12.** As Fig. 9 for location 9 (50.10354N, 14.45006E) — view of east facing wall.

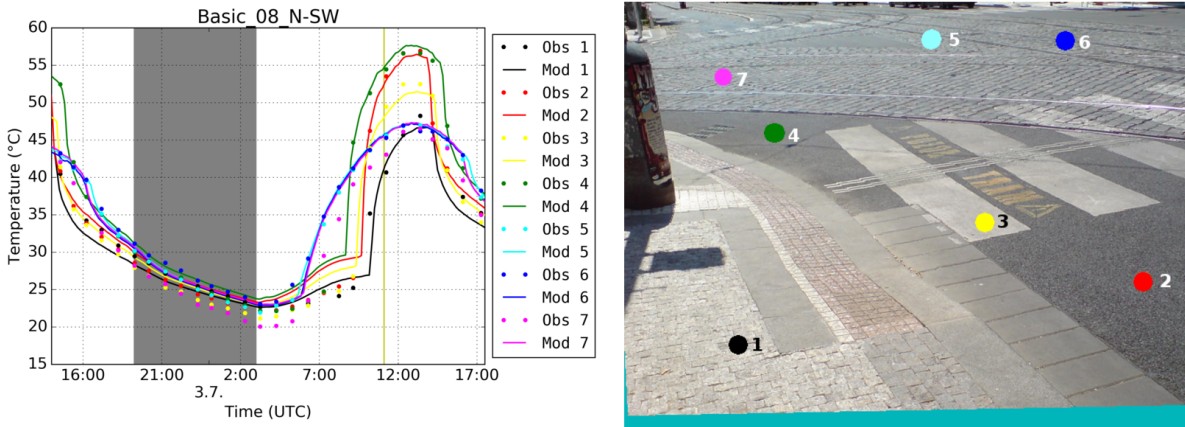

**Figure 13.** As Fig. 9 for location 8 (50.10340N, 14.45007E) — view of ground on the crossroads.

In Fig. 9, daytime temperatures of points 1 and 2 are captured quite well, while the model overestimates the temperature in point 3. In reality, this point is shaded by an alcove until 08:10 UTC (see the IR picture in Fig. S13) and thus it is irradiated approximately 1 hour later than point 1. As a consequence, the increase of its temperature is delayed and the reached maximum temperature is 4 °C lower than in point 1. This facade unevenness is not resolved by the topography model in PALM and it thus predicts the same values for points 1 and 3.

Figure 10 shows the same comparison for a west facing wall in the south arm of the north-south street measured from location 4. The temperature course in point 7 demonstrates the effect of surface shading by a tree that obstructs the solar radiation at this location between 13:10 and 14:50 UTC. This leads to a decrease in surface temperature between 13 and 15 UTC, whereas the surface temperature at the other points keeps increasing. This shading effect can also be seen in Fig. 12 for point 5, which is shaded by a tree between 06:15 and 08:15 UTC. Both cases are correctly represented by the model. Another illustration of tree shading is in Fig. S14.

The results for a north facing wall in the east arm of the west-east street are shown in Fig. 11. In contrast to other walls, where daytime temperatures are captured quite well, we can see that for north facing walls the model systematically overestimates the surface temperature (the same behaviour can be observed at another north facing wall observed from location 2, Fig. S5). This effect emerges when the opposite walls are fully irradiated by the sun (Fig. 11, 08:00–14:00 UTC). The same observation can be made for east facing walls in the afternoon hours (Fig. S8, 12:00–15:00 UTC). The possible reason for this overestimation is discussed later in Sect. 4.

Figure 12 shows the east facing wall of the north arm of north-south street. We can observe the effect of shading by opposite buildings here. As the sun rises and the shade casted by opposite buildings moves downward in the morning, the sun gradually irradiates point 3 (since 04:45 UTC), 2 (since 05:40 UTC), and 1 (since 06:10 UTC). This is reflected in observations and also in model results, although the modelled temperature in point 3 starts to increase somewhat later than the observed temperature

at the same point due to the discretized geometry of the buildings on the opposite side of the street. The effect of the shading of east facing walls during the sunrise is further visible in Fig. 15.

Finally, Fig. 13 shows the measurement of the ground surface temperature. The model captures the maximum values well, which are higher for asphalt (points 2, 3, 4) than for paving blocks (points 1, 5, 6, 7). The lower temperature of the white crosswalk, represented in the model as a one-grid-width belt with higher albedo, reflects in the model results as well. Also, the time when the temperature starts to increase in the morning is captured with some minor discrepancies, owing to the discretized representation of the surrounding buildings.

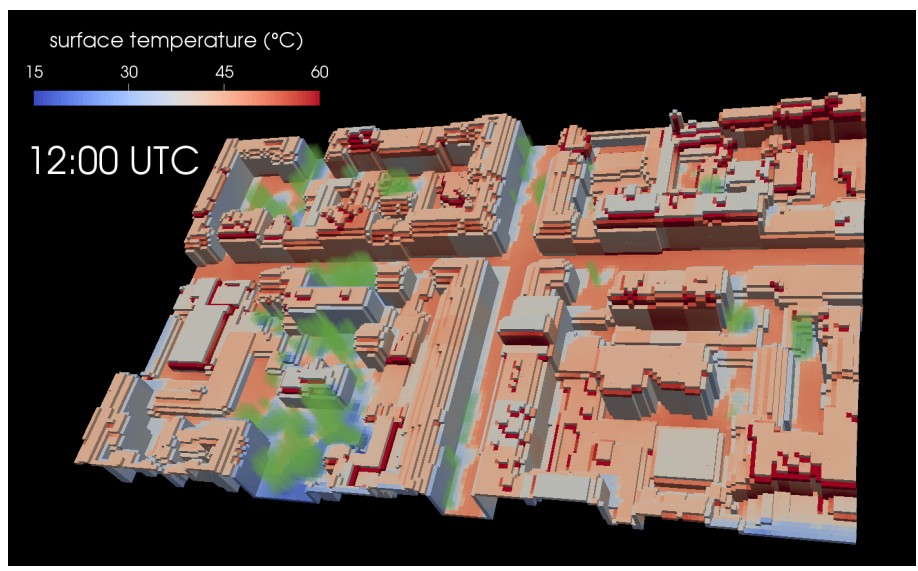

**Figure 14.** Modelled surface temperatures for the entire domain on 3 July at 12:00 UTC. Green areas represent vegetation (trees).

Figure 14 shows a view of the surface temperatures for the entire modelling domain on 3 July at 12:00 UTC, demonstrating different heating of facades due to different surface and material properties. As seen for all similarly irradiated surfaces (e.g. all south facing walls), the different wall properties lead to differently warmed surfaces. Further, cool spots resulting from shading by trees are clearly visible. This view also demonstrates the effect of transforming the real urban geometry into the regular modelling grid. The detailed view of east facing walls in the north-south street in the morning of 3 July is shown in Fig. 15. This picture shows surface temperatures after the sunrise at 06:00 UTC and 08:00 UTC. The view displays the effects of shading by buildings on the opposite side of the street as well as the thermal inertia of the material and the impact of different material properties.

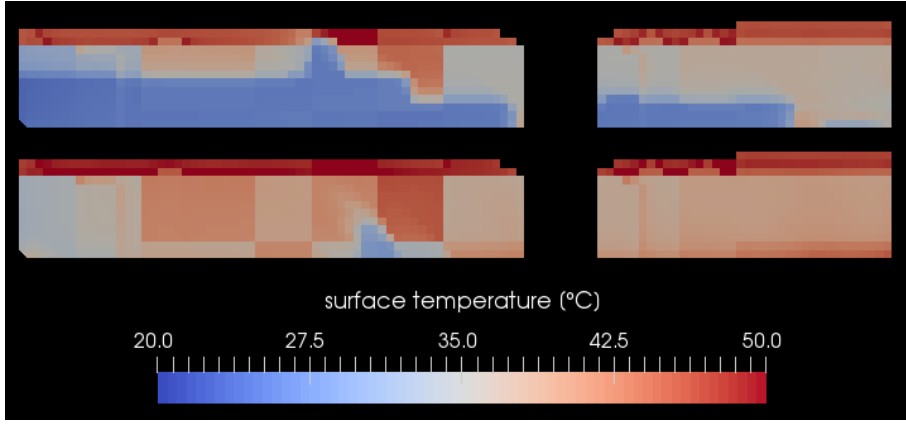

**Figure 15.** Modelled surface temperatures of the east facing walls in the north-south street after the sunrise on 3 July at 06:00 UTC (top) and 08:00 (bottom) UTC.

## 3.4 Sensitivity tests

### 3.4.1 Sensitivity to the dynamics of surface heat fluxes

In order to show the effect of dynamically calculated surface heat fluxes that are derived by the USM depending on given material properties, we performed another simulation with disabled USM (PALM noUSM). In order to make both simulations
comparable, they need to be based on the same energy input. This is achieved by prescribing a homogeneous surface heat flux to all surfaces (ground, roof, wall) in the noUSM setup, with this heat-flux value derived from the original USM simulation as the average over all surfaces in the entire domain in the target time. Simulation parameters in the noUSM simulation were set to values of the original USM setup from the selected period. The noUSM simulation ran for 5 hours to reach a quasi-steady state. Figure 16 shows the horizontal cross-section of the time-averaged (1 h) vertical velocity at 10 m above ground level
(AGL). Figures correspond to simulation time 3 July at 14:00 UTC, when west and south facing walls were fully irradiated and heated up by the sun. The wind above roof top was north-west and its strength was about $2 \mathrm{~m~s}^{-1}$. In the case PALM-noUSM, a typical vortex perpendicular to the street axis in the west-east oriented street and in the south part of the south-north oriented street was formed. In the reference case, however, the non-uniform heat flux was heating the air on the south and west facing walls, changing the strength of the street vortex. This effect is more intensely pronounced in the south arm of the north-south
oriented street where the strong vortex observed in case PALM-noUSM has significantly weakened. In Fig. 17 it becomes evident that the entire flow circulation pattern within this street arm had changed, leading to a change of the vortex orientation.

The accurate prediction of the canyon flow is an essential prerequisite, among others, for the accurate prediction of pollutant concentrations at street level and their vertical mixing. Our results — in line with previous studies — show that an interactive surface scheme is a crucial part of the urban modelling system and alters the canyon flow significantly.

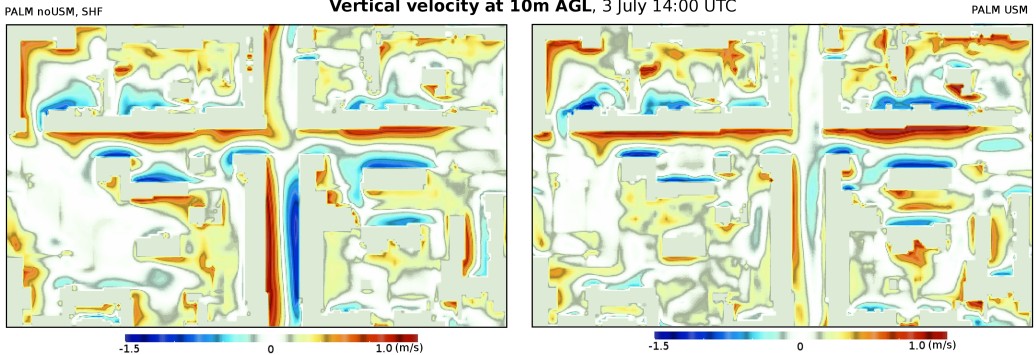

**Figure 16.** Horizontal view of modelled vertical velocity (one hour average) at 10 m AGL on 3 July, 14:00 UTC. Left figure presents stationary simulation without USM with constant surface heat fluxes and the right one the reference simulation with USM enabled.

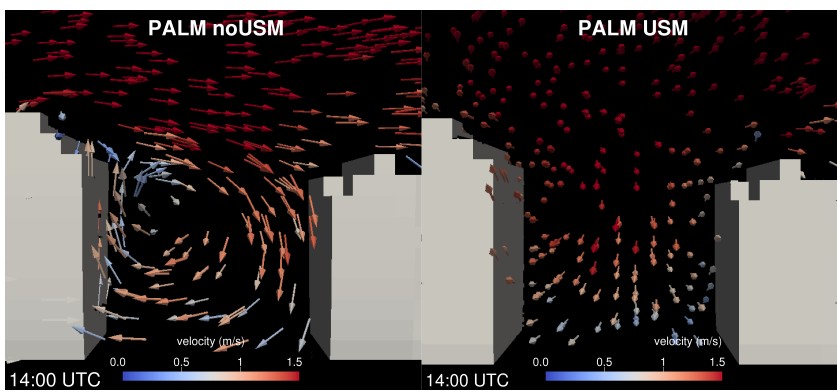

**Figure 17.** Wind field. Simulation without USM with prescribed surface heat fluxes (left) and with USM (right). View from the south border of the domain towards the crossroads.

### 3.4.2 Sensitivity to material parameters

Gathering of properties of individual materials and surface types is a challenging task. For our case, materials were categorized, and representative parameters were assigned to each category. The only exception is the albedo which was set individually for each particular surface based on surface colour. The uncertainty of the input parameters is high, though. In order to estimate related uncertainties of model results, a series of simulations was performed where one parameter was changed per simulation. The tests included the increase and decrease of albedo, thermal conductivity of both the material and skin layer, and roughness length of the surface. The albedo was modified by $\pm 0.2$ and all other parameters were adjusted by $\pm 30\ \%$ of their respective value. The sensitivities of the surface temperature at selected locations and evaluation points are presented in Fig. 18. The largest changes in model output are generally observed during daytime. The model behaves according to the physical meaning of the parameters: decreased roughness lowers turbulent exchange of heat between the surface and air, leading to the increase

of surface temperature when the air is colder, which is usually the case in our simulation. A decrease in roughness to 70 % leads to the increase of temperature by up to 4 °C. A decrease of thermal conductivity leads to more intense heating of the surface when the net heat flux is positive (usually during daytime) and to less intense cooling when it is negative (usually during night). The decrease of the albedo leads to higher absorption of SW radiation and an increase of the surface temperature during daytime. The sensitivity of the modelled surface temperature can reach up to 5 °C.

This sensitivity analysis shows that even moderate changes in the wall material properties can lead to differences of the surface temperatures of a few degrees Celsius. As mentioned in Sect. 3.3, we observe an overestimation at some evaluation points located at the lower parts of the buildings (location 3, point 4; location 4 point 1, location 5 point 1). We hypothesize that a possible reason can be that walls at lower parts of buildings can be built from different material than the upper floors. In that case the thermal conductivity used in the model would be different than its true value. This can be the possible explanation of some discrepancies between model and observation. On the other hand, some discrepancies (e.g. overestimation of temperatures on north facing walls) are systematic and they can be probably attributed to some limitation of the setup, or the model itself. Some possible reasons are discussed in Sect. 4.

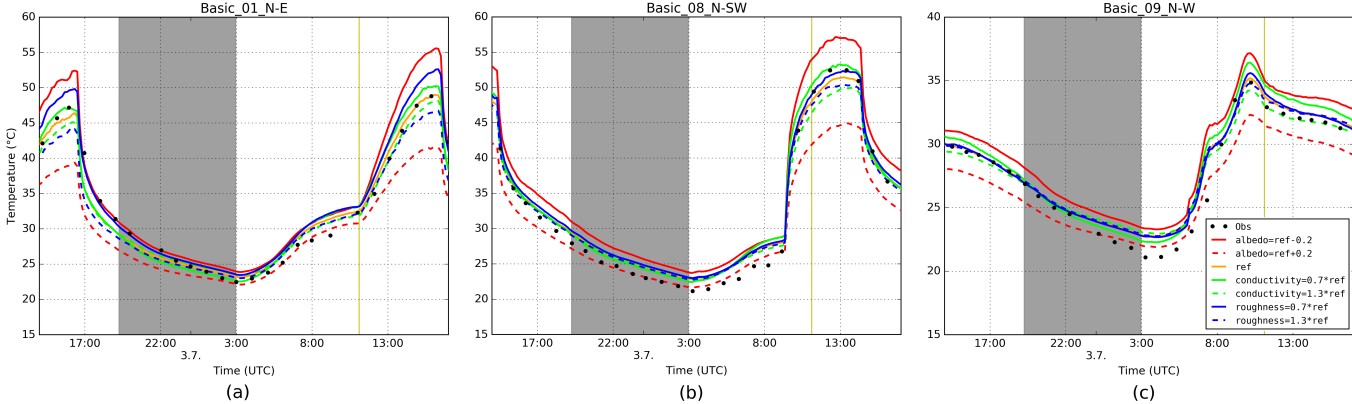

**Figure 18.** The sensitivity of PALM-USM to changes of surface and material parameters: (a) for the west facing wall measured from location 1 for evaluation point 1, (b) for the ground measured from location 8 for evaluation point 3 and (c) for the east facing wall measured from location 9 for evaluation point 5. *ref* is the reference run, other lines correspond to the increase (dashed line) and decrease (solid line) of respective parameters with respect to the reference run: *albedo* was increased/decreased by 0.2; thermal conductivity of materials and skin surface layer (*conductivity*) was increased/decreased by 30 % and so was the roughness length (*roughness*). *Obs* is the measurement in evaluation point.

## 4 Discussion

The deficiencies in the model's description of reality and the discrepancies against observations may arise from limitations, simplifications, and omissions within the model itself and from limited exactness, representativity and appropriateness of the model setup.

### 4.1 Limitations of the model

The USM and its radiative transfer model (RTM) assumes only a diffuse reflection and does not treat windows. In our configuration, specular reflection can play a role for glossy surfaces like flagstones and glass. Windows also transmit part of the radiation into the building and only the rest is absorbed by the glass itself. Examples of IR camera observations of the south facing wall taken from location 6 at 01:08 UTC, 10:08 UTC and 13:17 UTC are shown in Fig. S15. The effective temperature of the window is usually lower by several K to a few tens of K compared to the surrounding wall. Since this is not captured by the model, the longwave radiative flux emitted from those walls can be overestimated. The approximate calculation for the area captured by these IR photos shows that average heat fluxes of the wall with windows are lower by $7 \text{ W m}^{-2}$ (01:08 UTC), $27 \text{ W m}^{-2}$ (10:08 UTC) and $25 \text{ W m}^{-2}$ (13:17 UTC), as compared to the wall without windows. The difference in average effective temperature of the wall is 1.5 K, 4 K and 3.5 K, respectively. As mentioned in Sect. 3.3, the model overestimates the temperature of north facing walls. Since the modelled net radiation on the north facing walls ranges from $-20 \text{ W m}^{-2}$ during night up to $70 \text{ W m}^{-2}$ during the day, and since the opposite walls represent about one third to half of the visible area, these differences are not negligible and may be responsible at least partly for the overestimation of the surface temperature of north facing walls. This suggests that an extension of the USM by a proper window model is very desirable.

The RTM also simulates only a finite, configured number of reflections. After that, the remainder of reflected irradiance is considered fully absorbed by the respective surface. The amount of absorbed residual irradiance is available among model outputs and it can be used to find an optimum number of reflections until the residual irradiance is negligible. The optimum setting depends on albedo and emissivity of the surfaces. In our setup, the residual irradiance was below 1 % of surface's total at most surfaces after three reflections and it was negligible after five reflections.

The current version of the RTM model doesn't simulate absorption, emission and scattering of radiation in the air within the urban canopy layer, thus, it is not suitable for modelling of situations with extremely low visibility like fog, dense rain or heavy air pollution. However, under clear air conditions and in an urban setup where typical distances of radiatively interacting surfaces are in the order of metres or tens of metres, these processes are negligible[8]. Most of the solar radiation's interaction with the atmosphere happens on the long paths from top of atmosphere to ground and during the interaction with clouds, i.e. above the urban layer, where the method of modelling of these processes depends on the selected solar radiation model in PALM.

---

[8]Using MODTRAN (Berk et al., 2014) for a clean-air summer urban atmosphere, transmissivity for 10 µm radiation (i.e. peak wavelength of black-body radiance at 300 K) per 1 km of air is approximately 0.85, which equals 0.998 per 10 m.

USM is currently coupled with PALM's simple clear-sky radiation model, which provides only limited information on sky longwave radiation and it does not provide air heating and cooling rates. This limitation will be overcome in the near future when the USM will be coupled to the more advanced RRTMG model in PALM.

Shading by plant canopy is only modelled for shortwave radiation; in the longwave spectrum, the plant canopy is considered fully transparent. Typical daily maxima of SW radiative fluxes (mostly from direct solar radiation) are much higher than maxima of LW fluxes. Moreover, much of the LW heat exchange is compensated when surfaces are near radiative equilibrium. Therefore, for the LW shading by plant canopy to cause significant changes in the heat fluxes, two conditions must occur simultaneously: the plant canopy and the affected surface has to occupy a large portion of each other's field of view (e.g. a large and dense tree close to a wall); the temperature of the plant canopy, the affected surface and the background field of view has to differ significantly (e.g. the wall is under direct sunlight and the plant canopy is shaded or cooled by convection).

To illustrate the amount of affected heat flux, let us propose a simple realistic scenario where these effects are very strong. Let us have two opposing walls, each occupying 50 % of the other's field of view (without regard to plant canopy) and let us add a row of trees directly between the walls, blocking 30 % of the mutual radiative exchange between the walls. Let the temperature of the ambient air, one of the walls and the plant canopy be 300 K and let the other wall heat to 320 K due to strong direct sunlight.

Under these conditions, the cool wall would be receiving 68 W m$^{-2}$ of excess total radiative flux (absorption minus emission) due to the opposing wall being hotter. The hot wall would be losing the same excess total flux due to the opposing wall being cooler, both when not accounting for shading by plant canopy. With the plant canopy, the cool wall would only be receiving 41 W m$^{-2}$ of excess total flux from the opposing wall and the remainder flux of 21 W m$^{-2}$ would be absorbed by the plant canopy. The hotter wall would experience the same radiative cooling as without plant canopy.

With regard to our test scenario, we accepted the simplification, considering that the demonstrated omission would affect only a few spots in the modelled domain. Modelling of LW interaction with plant canopy is planned for the next version of the model.

Plant leafs are treated in the USM as having zero thermal capacity and a similar temperature to the surrounding air. Any radiation absorbed by leafs, directly heats the surrounding modelled air mass. In reality, plant leafs are thin, they have a large surface area and they readily exchange heat with air. This simplification is common among radiative transfer models (see e.g. Dai et al., 2003) and it is also in accordance with the current implementation of the non-urban plant canopy model in PALM.

Evapotranspiration of the plant canopy is not modelled and surfaces are considered impervious to water. Generally, these are important processes which will be accounted for in the upcoming versions of the model. The importance of evapotranspiration and latent heat grows with the modelled proportion of vegetation. The measurements in Grimmond and Oke (1999) - specifically in Table 2/page 925 - show that in their case, latent heat flow vs. net radiation ratio ranged from 4 % in the downtown area (similar to the streets in our test case) up to 37 % in the suburban area with a high fraction of vegetation. In our case, however, street surfaces are covered by asphalt or granite paving blocks with gaps filled by asphalt. Only a part of the pavements in the north-south street that are paved with limestone blocks can be considered pervious to water to a larger extent. There are only few trees in the streets, concentrated mostly in the north-south street, and their treetops are not very dense (Fig. 12 and

10). Moreover, the last precipitation before the observation campaign was recorded in Prague on 29 June ($0.3 \ \mathrm{mm \ day}^{-1}$). Therefore, it can be expected that the neglect of these processes will not have large effects on the evaluation presented in this article.

## 4.2 Appropriateness of the presented setup

One of the potential issues of the setup is the model domain's horizontal size. The CBL height reached values of up to 2000 m during daytime (see Fig. 7). It is well known that the largest structures in a CBL scale with the height of the boundary layer, and they typically form hexagonal cellular patterns. In this context, the chosen horizontal model domain is too small to resolve these structures. We must thus expect that the largest turbulent eddies were not able to freely develop during daytime. Nevertheless, the feedback of these eddies onto the surface-subsurface continuum can be regarded to be small. This is supported by our recent experiences using the PALM-LSM system for a dry bare-soil configuration (work in progress, not shown). Moreover, as we have seen in Sect. 3.3, the simulated skin temperatures compare well with observations and do not display significant fluctuations at turbulent time scales.

However, we have seen that the vertical profiles of potential temperature display an unusual unstably-stratified shape during daytime. To estimate the possible influence of the domain extent on the mean vertical profiles, an idealized simulation was performed. The horizontal grid size was increased by a factor of ten to 20.8 m, while all the rest of the setup was kept unchanged. This artificially increased the simulation domain to a horizontal extent of 3760 m × 2260 m, without having to increase the number of computational grid points. The topography of this domain is not fully comparable with the original domain as the street width is ten times larger and also the ratio of the sizes of individual grids differs from the original setup. Nevertheless, it can indicate the overall characteristics of the setup with a large domain. Vertical profiles of potential temperature from this simulation are shown in Fig. 19. Compared to Fig. 7 we can see that the unstable stratification that was observed at 12:00 UTC and reached up to 1.5 km changed to near-neutral conditions in the large-domain simulation, corresponding to well-mixed boundary layer conditions. Based on this finding, we must be aware of the fact that the formation of the largest turbulent eddies was inhibited by the small model domain, but that they are essential for the efficient vertical transport of heat. While this is no problem for RANS-type models, where all turbulent eddies are parameterized, it imposes a limitation for the application of LES models in urban areas, as one has to ensure that the horizontal model domain is at least of the size of the boundary-layer depth and thus will require enormous computational resources. However, one of the methods to overcome this problem is already under way. This is the two-way nesting system for urban applications that has already been implemented in PALM and allows for using finer grid spacings in areas of special interest while having coarser grids in the surroundings.

Another limitation of the setup is the fixed indoor temperature. The studied simulation spans only over two days with similar summer weather conditions. Considering the heat capacity of walls, the influence of changes of the indoor temperature can be regarded small for the presented simulation. This issue could be more important for long term simulations.

Gathering information about the detailed structure of the walls in the entire domain constitutes a significant challenge. For practical reasons, the material of every particular wall is considered homogeneous in our simulation. Thus, the thermal conductivity of the sandwich structure of insulated walls is not well represented, as well as the structure of pavements and

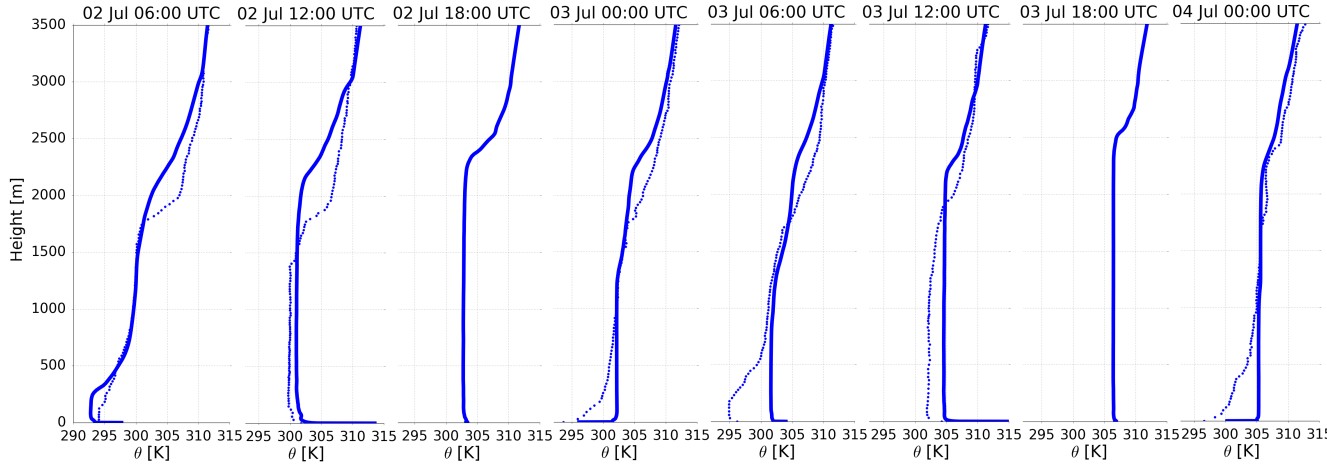

**Figure 19.** As Fig. 7 for the idealized simulation with enlarged domain (see Sect. 4).

streets. The conductivity of the entire wall structure can be substituted by the average conductivity with no effect to the long-term average heat flux going through the wall. However, layers of walls described by the same average parameters can actually have different thermal capacities and conductivities, resulting in different thermal dynamics. This can partially influence the shape of the surface temperatures in affected places.

Anthropogenic heat sources were limited to traffic in our test case. Other potentially significant anthropogenic source can be heat emissions from buildings. Considering that the period of interest is in the summer, the relevant processes would be air conditioning, which however is not common in this part of the city.

When evaluating model results, the uncertainty of the measurements must also be taken into account. In our case, the measurements are affected by not using emissivity correction option of the IR camera, and by possible reflections, mainly
in the case of horizontal surfaces. For this reason, only those points, where the influence of reflections was supposed to be negligible, were considered for evaluation.

The presented evaluation of the model is limited to a specific city district and meteorological condition. However, since the model is based on general formulations, it should be applicable for arbitrary configurations of fully-urbanized areas. Concerning the meteorological setup, we suppose that the model with its current limitations is yet suitable to represent the urban
surface-atmosphere interactions under meteorological conditions where the omitted processes do not play a significant role. Model limitations will be resolved in the future PALM-USM versions, which will be validated accordingly for a wider range of meteorological conditions and surface types. A key factor for a successful validation is the good knowledge of these conditions and the properties of all urban surface elements.

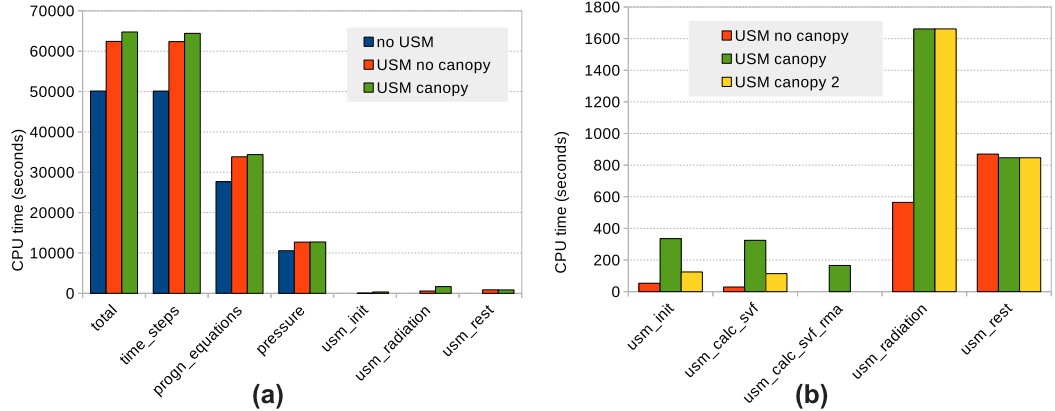

**Figure 20.** Comparison of duration of the model run and time spent in chosen subprocesses of the model (a) and detailed comparison of parts of USM model (b). Meaning of data series: "no USM" the run of PALM with USM switched off, "USM no canopy" the run with USM with no plant canopy, "USM canopy" run with USM and plant canopy, "USM canopy 2" the same run with the model configuration option usm_lad_rma turned off. Meaning of items: *total* – total CPU time of the model run, *time_steps* – time spent in time stepping, *progn_equations* – evaluation of all prognostic equations, *pressure* – pressure calculation, *usm_init* – initialization routines of USM, *usm_radiation* – calculation of USM radiation model, *usm_rest* – remaining USM processes (particularly energy balance and material thermal diffusion), *usm_calc_svf* – calculation of SVF and CSF, *usm_calc_svf_rma* – time spent with one-sided MPI communication. The setup of the model corresponds to the setup described in Sect. 3.2 with reduced number of layers to 81.

## 5  Computational aspects

The correct functionality and computational efficiency of the implementation of USM was verified in various environments. The tested configurations varied in processor type (Intel[9], AMD[10]), compiler (GNU[11], PGI[12], Intel[13]), implementation of MPI (MVAPICH2[14], IMPI[15]), and other aspects. The comparison presented in this chapter was performed on the Salomon supercomputer[16] with Intel C and Fortran compilers and Intel MPI (2016 versions for all). The setup of the model corresponds to the setup described in Sect. 3.2 with decreased number of vertical layers to 81 and one-day simulation extent to get reasonable computational time also for smaller number of MPI processes.

Figure 20a shows the comparison of the total CPU time of the model run and the CPU time needed for particular chosen subroutines. Almost all of the total time is spent on time stepping. The direct expense of the USM can be split into three parts: the time spent in initialization routines of the USM at the start of the model run, the time needed for calculation of the urban

---

[9]https://ark.intel.com/

[10]http://www.amd.com/en-us/products/processors

[11]https://gcc.gnu.org/

[12]http://www.pgroup.com/

[13]https://software.intel.com/en-us/intel-compilers

[14]http://mvapich.cse.ohio-state.edu/

[15]https://software.intel.com/en-us/intel-mpi-library

[16]https://docs.it4i.cz/salomon/introduction

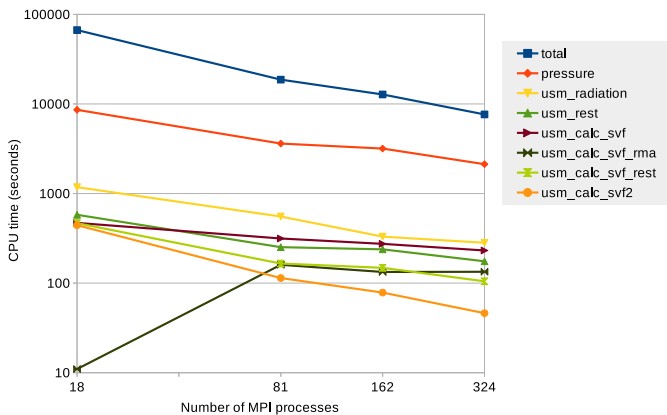

**Figure 21.** Comparison of the calculation time spent in model PALM-USM and in its chosen parts for various number of cores. Model configuration and the meaning of items is the same as in Fig. 20, additionally *usm_calc_svf_rest* shows the difference *usm_calc_svf* − *usm_calc_svf_rma* and *usm_calc_svf2* depicts *usm_calc_svf* in case of the run with option usm_lad_rma set to false.

radiation model and finally the time of remaining USM processes, particularly the energy balance and the material thermal diffusion. The total increase of the calculation time with USM switched on is about 25 % (29 % with plant canopy). However, the direct USM calculation cost presents only about 2 % (4 % with plant canopy) of the total calculation time. The rest of the increase can be attributed to the raised turbulent flow which results in decreased time step. Figure 20b shows the detailed

comparison of USM processes. The initialization time of the USM is dominated by the calculation of SVF and CSF and about half of this calculation is spent with one-sided MPI communication in case of the run with plant canopy. The utilization of one-sided MPI routines can be avoided by distributing the global leaf area density (LAD) array into all MPI processes by setting the model configuration parameter usm_lad_rma to false, which reduces the time spent in USM initialization process and markedly improves the scaling behaviour.

The effectiveness of the parallelization has been tested for number of MPI processes in range from 18 to 324 for the simulation length 24 hours and the results are shown in Figs. 21 and 22. Figure 21 compares the CPU time needed for calculation of whole model PALM-USM and its chosen individual parts. Figures 22a and 22b show the effectiveness of the parallelization relative to a run with 18 processes for simulation with and without calculation of plant canopy, respectively. The graphs suggest that time-stepping routines *usm_radiation* and *usm_rest* scale similarly to calculation of the pressure which is the most time

consuming individual process of the PALM model. The calculation of SVF during initialization phase scales excellently in the tested range according to Fig. 22b. Scaling of the calculation of CSF is on a par with the whole model PALM for configuration with the LAD array distributed into all processes (Fig. 22a, item *usm_calc_svf2*) while scaling of the *usm_calc_svf* is limited by latency of one-sided MPI operations implemented by an Infiniband RMA backend (Fig. 21, *usm_calc_svf_rma*). (Note that the run with 18 processes fits into one node of the computational cluster and all MPI communication is done through a shared

memory backend in our setup.) On the other hand, it also suggests that the computation of CSF can scale well when the computational domain extends. However, the testing domain is relatively small and additional tests with larger domains are needed

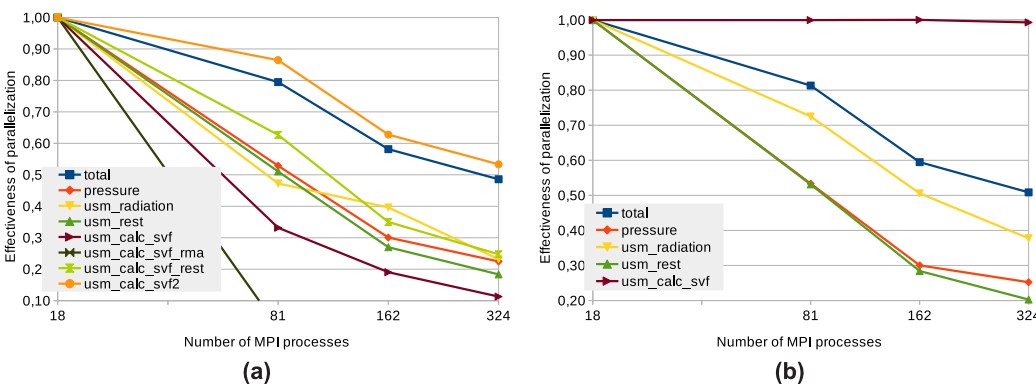

**Figure 22.** Effectiveness of parallelization of chosen subroutines: (a) simulation with plant canopy, (b) without plant canopy. Meaning of the items is the same as in Fig. 21.

to extract deeper insight into performance and scaling of PALM-USM. The first tests with the domain extent over 3 km suggest that the scalability of the present version of RTM is limited by growing memory requirements, particularly by the growth of the number of SVF and CSF. This issue will be solved in the next version of RTM.

## 6  Conclusions

The new model of energy processes in urban environments was developed and integrated into the PALM model as an optional module. The USM utilizes meteorological values calculated by PALM, and it provides sensible heat fluxes as boundary conditions for the atmospheric flow. In this paper, we described the technical details of the USM formulation. Moreover, a first evaluation against data from a measurement campaign in Prague, Czech Republic was performed, as well as basic sensitivity tests to material parameters. The results are generally in good agreement with observations for our test case. In particular, the
evaluation incorporated a detailed comparison of the simulated building-wall and street surface temperatures with IR camera measurements. The results showed that the diurnal variation of the surface temperature was very well captured except for the north facing walls, where the temperatures were overestimated by up to 3 °C. A likewise overestimation was also found on some other walls during nighttime hours. These differences can be attributed to inaccurate description of the urban parameters such as heat capacity and conductivity of the wall material as well as to some limitations of the current version of the model
and model setup, e.g. no window model implementation.

Uncertainties due to the sensitivity to the setting of material parameters were estimated in a suite of simulations altering three basic parameters: albedo ($\pm$ 0.2), thermal conductivity and roughness length (both $\pm$ 30 %). The results show that the tested albedo variation generally induces the largest changes in surface temperatures (up to $\pm 5$ °C). The overall magnitude of changes confirms that the proper setting of material parameters is crucial for the application of the model in real-case simulations.

For tested configurations, the USM shows very moderate computational demand in the context of the other PALM components.

Addressing the current limitations of the USM is a subject of current and future development inside the PALM community. Major changes to the current USM version will involve the implementation of a tile approach to account for windows and green roofs/walls. An energy balance solver for trees will be added in order to explicitly predict the turbulent fluxes of sensible and latent heat from leafs. Also, the wall model will be coupled to an indoor climate and energy demand model, which predicts the indoor temperature, but also the energy demand of the buildings, including anthropogenic waste heat emissions from the buildings due to heating and air conditioning. Furthermore, the scalability of the urban radiative transfer model will be rigorously enhanced to allow for using larger computational grids. The USM will be coupled to the RRTMG radiation model to improve the radiation input at the top of the urban layer. Finally, the USM will be coupled with the PALM-LSM, which allows to represent processes related to latent heat (evaporation, transpiration) and urban areas that exhibit larger areas of parks and pervious surfaces compared to the present test case. Many of these actions are work in progress within the framework of the project MOSAIK.

Despite the current limitations, the PALM-USM model provides a new useful tool for climatology studies of urbanized areas, and has been successfully used to simulate urban development scenarios for the city of Prague.

*Code availability.* The USM code is freely available and it is distributed under the GNU General Public License v3[17]. Its source code is a part of PALM and it can be downloaded from the PALM web page[18] via the SVN server[19] since PALM revision 2008. The particular version used for computation of the simulations presented in this article is available in the branch "resler", revision 2325. This branch version includes also a simple air pollution model.

*Competing interests.* The authors declare that they have no conflict of interest.

*Acknowledgements.* Authors would like to thank two anonymous reviewers for the comments that helped to improve the manuscript considerably. We acknowledge following projects that supported this research. This work was done within the UrbanAdapt project (EHP-CZ02-OV-1-036-2015) supported by grant from Iceland, Liechtenstein and Norway[20]. This work was also supported by the long-term strategic development financing of the Institute of Computer Science of the Czech Academy of Sciences (RVO:67985807). Some of the simulations were done on the supercomputer Salomon, which was supported by The Ministry of Education, Youth and Sports from the Large Infrastructures for Research, Experimental Development and Innovations project IT4Innovations National Supercomputing Center – LM2015070. Coauthors B. Maronga and F. Kanani-Sühring are funded by the German Federal Ministry of Education and Research (BMBF) under grant

---

[17] http://www.gnu.org/copyleft/gpl.html

[18] https://palm.muk.uni-hannover.de

[19] http://subversion.apache.org

[20] http://urbanadapt.cz/en

01LP1601A (project MOSAIK[21]) within the framework of Research for Sustainable Development (FONA[22]), which is greatly acknowledged. Authors would like to thank Linton Corbet for language revisions and useful comments. We also would like to thank to the coordinator of the UrbanAdapt project Global Change Research Institute (CzechGlobe) for lending IR camera and František Zemek for his help with ob-servation campaign. We thank to the UrbanAdapt project partner Prague Institute of Planning and Development for providing geographical

5    data and also to the ATEM company for its help with the data processing.

---

[21]http://uc2-mosaik.org

[22]www.fona.de

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
