# Peer review of "A new urban surface model integrated in the large-eddy simulation model PALM (PALM-USM v1.0)"

_Geoscientific Model Development, 2017_

## Referee Comment (RC1) · Anonymous Referee #1 · 11 Apr 2017

**General comments**

The manuscript *A new urban surface model in the large-eddy simulation model PALM* by Resler et al. describes the addition of new radiation and energy model (USM) to the existing large eddy simulation model PALM. This is an important addition to PALM, which before did not account for radiation transfer in complex street canyons well. This is important for several urban processes and thus the new model clearly is a welcomed addition to the capabilities of PALM. There are however some challenges, what comes to representativeness of the new module and presentation of the methods and results (see below), in the manuscript that needs to be addressed. Also the language of the manuscript needs further improvements. There are several parts that need revision and in the minor comments I've tried to point out some of them, but I suggest the authors to go through the language once again before resubmitting the manuscript.

[Figure]

Thus I suggest major revisions to the paper before it can be accepted for publication to Geoscientific Model Development.

**Major comments**

*Representativeness of the new module*
To my understanding the anthropogenic heat emissions only from traffic are accounted for. This might not be an issue in summer at the site of evaluation as there are no heat emissions from buildings, but what if the model evaluation would have been made in winter or in other city with huge need for air conditioning? This is a clear lack in the USM as the authors could have implemented e.g. a simple temperature related anthropogenic heat emission model following some activity profile similar to the traffic heat emissions. Also, USM neglects latent heat flux component from the surface energy balance, which can be important in neighbourhoods with more vegetation. At the same time I understand why in the first step of the model development only the some (most crucial?) points of the complex energy system are included, but the authors should still comment in more detail about the limitations of the model. Some limitation is currently described on P10, but the representativeness and limitations of current USM model version should be described in detail either in the results/conclusions or in a separate section after the results.

On P6, L3-10, the authors list radiation-related processes that were omitted from the radiation model. Could the authors add what is the level of impact these omitted processes might have on the model performance?

*Surface properties in USM*
Please add somewhere to Section 2.3 that the needed surface properties to run USM are given also in the Supplementary material. How is the clearness index (P5, L26) given to the model?

For the evaluation part, it would be important to know what surface property values were used for the different observational points presented in Figs. 7-11. Maybe a table

to the main paper or supplementary material would work? Also in the results the effect of the different properties could be extended. What was the anthropogenic profile you used for traffic emission? It would be good to plot this together with traffic rates and meteorology for the case study period (see comment below). The obtained traffic heat emissions (P13, L19-20) are rather large during peak traffic hours. To me they seem unrealistic so could the authors comment how they compare with other studies.

*Model runs*
The vertical domain height is high when compared to the horizontal scale of the simulated area. At the same time the authors say that outside domain area has minor impact on the processes within the modelling domain, but such a high vertical domain makes me doubt this. This must be affected by some further away surface not with similar characteristics as the study area. Could the authors comment this?

It is not explained clearly why did the authors use WRF data to provide forcing for the run. This is shortly described in the results (P15, L4-5, 7-8) but the explanation should be given already in section Model setup. How did the model forcing data look relative to the Karlow station data? Air temperature data is given in Figure 8, but how did wind look like? suggesti that new figure where meteorological variables from WRF and observations (Tair, wind, solar radiation) and traffic rates for the simulated period would be plotted.

*Results*
The model evaluation section is currently quite poorly written and needs revisions. Text on P15, L13-19 is unclearly written and jumps between differences in the observation points, comparison between model output and observations and furthermore locations. Also, is this part referring only to location 1 as its not clear from the text. If yes then the general conclusion that modelled wall temperature drops faster after sunset is not valid as only on half of the points this is the case and in half not. Rather this pace of cooling could be related to thermal properties of the different points. Please rewrite.

On P17, L4, the overestimation takes place only in daytime. Please add this information. The authors mention here that the daytime overestimation could be due to heat capacity of the wall. This could be the case indeed as it seems that the surface is not storing enough energy in daytime and release it enough in night-time. This should be discussed more properly in the results section.

In generally more text about the surface properties and their impact to the model performance should have been added. I'm missing some sensitivity tests about the impact of surface properties to the model performance. For example the authors could choose some location point from Figs. 10 and 11 where the surface properties would be slightly changed and improve relative to the observations. This particularly in the case of Figure 11, where the surface temperatures seem to be completely off.

The authors could add more analysis on section 3.5 about the differences between PALM without and with USM. How great impact does the addition of USM have on the turbulent mixing. Could maybe some spatial means at different heights be calculated to really see how mixing is improved? Or maybe showing vertical profiles from certain points on the main streets? Due to missing measurements, I guess the authors cannot really comment is the representation of turbulent mixing improved or not.

The problems related to model/observation comparisons are not mentioned in the conclusions. Possible needs to improvements should be added there.

**Minor comments**

P1, L1: "a direct effect" -> "direct effects"
P1, L2: "This implies the need for a reliable tool for climatology studies that supports urban planning and development strategies" -> "This implies that reliable tools for local urban climate studies supporting sustainable urban planning are needed"
P1, L4-5: "...a new Urban Surface Model (USM) describing the surface energy processes for urban environments was developed..."
P1, L7: In the model the authors neglect latent heat flux and thus are not calculating

the total energy balance for impervious surfaces. Please reword here.

P1, L9: Please open what MPI means.

P1, L19: I would remove the first sentence: it is said in the abstract already.

P1, L20: Add "in future" after increasing, add "local urban" in front of climate.

P2, L2-3: I would reword this difficult sentence as e.g. it is not clear what is meant with "sound scientific background". I guess the authors mean rather tools?

P2, L4: Should be "...phenomenon related to...". I would change the UHI reference to the original paper by Oke.

P2, L6: "...retention energy of urban surfaces and increased heat emissions from human activities."

P2, L7: Not only building shadows create cool islands but also tree shadows and increased evaporation.

P2, L7-L27: In these lines there is unnecessary repetition and should be restricted. After the cool island should be the whole description how the heat islands are commonly studied and after that what problems these methods meet so that eventually LES modelling is required to understand the issue. Also on some lines the authors talk about urban processes generally and on some lines only on the urban heat island. Also the references on L17 consider only UHI and not e.g. air quality that the authors mention on L2 at the same page.

P2, L33: can be -> is

P3, L3: Remove comma from the front of LES.

P3, L3-4: "Many of the CFD models do not contain appropriate radiative models and to overcome this deficiency, an independent radiative models with the resulting radiation fluxes have been imported into the CFD model..."

P3, L9-16: The objectives of the manuscript focus now on the project under which the project is made of, but these should be rephrased to be more general and representative for the actual study. At the same time LES does not require CFD in its front so please remove it.

P3, L18: Abbreviation PALM should be opened in the text here.

P3, L28: "...obstacles as well as the landform"

P3, L31: Replace next with Secondly; "...radiative exchange at the surface..."

P4, L2: of using -> to use

P4, L7: "...PALM-LES, further extends the surface parameterisations..."

P4, L12 "...plant canopies have not been..."

P4, L16: Again only radiation and direct heat flux is considered: not the whole energy balance

P4, L19: as well as -> and; material -> materials

P4, L24: heat fluxes -> sensible and storage heat fluxes. Also I would add already here at the end of the sentence that heat consumed to evaporation is not accounted for.

P4, L25: "The energy budget in the skin layer...". The reference to PALM-LSM is not needed here again.

P4, L26: Anthropogenic heat flux is missing from the equation. Units are missing from the variable descriptions (an also from later equations). Please add throughout the manuscript. P5, L3: Why here the potential temperature is used whereas in Equation (1) there is air temperature? Shouldn't zero refer to skin surface and not surface?

P5, L4-7: Could the authors add a bit more information about the parameterizations especially as the Maronga and Bosveld paper has only been submitted.

P5, L13: Should be systematically PALM-LSM.

P5, L16: You can replace "Ground heat flux" with "G".

P5, L17: Following equation 1 the layer next to surface should be skin layer?

P5, L22: The title could be "Multi-reflection transfer model" as then it would be systematic with the text on P4, L20-22.

P5, L29-L30: The processes related to shortwave radiation is unclear. It is written that process "Radiation sources from the sun...using the relative position of the sun" is modelled, but from the above text I get the impression that the shortwave radiation on top of the canopy is obtained from the chosen radiation module in PALM. Thus, please be more specific here.

P6, L12: Here the authors use word irradiance at each surface whereas in the energy balance equation (1) they use net radiation. Please, systematize throughout the manuscript.

P6, L16: Remove "also" from the sentence

P6, L20: The abbreviation for the differential view factor (uppercase d should be give here).

P6, L21: As this is generally the equation used for sky-view factor I would add a reference to the equation.

P6, L23: The separation distance is explained in the previous sentence on the same line and thus abbreviation s can be used after "Under the assumption..."

P6, L26: Please explain what A' means. In generally text and equations are not very clear starting from here and ending on P7, L2 and additional information source needs to sought if you are not that familiar with the calculation of view factors. Thus I suggest the authors to add a bit more explanation to this part of the manuscript with proper description of the variables used in the equations.

P6, L8: Same applies to Equation (5). It is not explained that this equation is valid for the case where two canopy grid boxes C and D are between surfaces A and B. First it should be given what is the RCSF for a single grid box C or D.

P10, L16-28: The order of explanation is strange here. The authors first describe the measurement locations before explaining what instruments are used. I would suggest to explain first what is measured and how (surface temperature using infrared camera) and then the actual locations of the measurements. How far was the camera from the surfaces and what was its view in degrees.

P11, L22: What is meant with "slight changes in camera position"?

P11, L4: It would be better to describe here the selected surface cover types and not in the results section.

P12, L5-6: How was air temperature measured?

P12, L7-12: It would be nice to have the meteorological conditions plotted in a Figure from the around 1.5 day measurements campaign (see major comments)

[Figure]

P12, L29: What is Medard prediction system?

P13, L24-25: It would be nice to have these times in the meteorological figure as lines or as radiation itself.

P13, L26: Add "modelled surface temperatures"

P13, L27: It is better not to use the name of the street when referring but rather use "along the west-east street"

P14, L4: "...of the domain to illustrate the effects of tree..."

P15, L1: I would remind here what kind of measurement location location 1 was.

P15, L2: What is meant with indicative measurement? The automatic weather station is not mentioned in methods and thus should be added there.

P15, L3: Klementinum complex does not say much to the reader. Is this large area? Is the station part of official meteorological monitoring? This should all be added to the methods. P15, L5: I would simplify the sentence: "The street level air temperature form PALM-USM is in..."

P15, L7: "...temperatures..."

P15, L9-10: "Comparisons...are displayed..."

P15, L10: "...observed temperature patterns..."

P15, L26 onwards; I would move Figure 5 to Supplementary material as there are already many figures, and new should be added. The figure is nice looking but not relevant for the actual paper.

P18, L2: plays -> play. Please open here what do you mean by these effects.

P22, L1: had been -> was

P22, L1: "...in the range..."

P23, L7-8: only sensible heat flux is given.

Figure 2: Add scale also to this aerial image similarly to Fig. 3. You could also draw the area of the observational image to Fig. 3 in a similar fashion as you show the modelling area with green.

Figure 4: Figure text needs more explanation: It would be good to add the date and that the data shown is modelled.

Figure 5: Scale is missing from the figure. Would it be possible to add this small area to Fig. 3 as a box?

Figure 6: It is difficult to see the lines if printed in black and white. Some of the darker colour lines could be plotted as dashed lines to separate them. I think AGL is not explained in the manuscript. Also as the figure should be able to be looked without references to the text, the authors should add the location 1 above road to the figure text as well as the time (2-3 July 14:00-17:00).

Figures 7-11: The figure texts are very poor currently. Please modify them to include the day, time period, and on the first one also description about the solid and dotted lines. Also in Figure 7 it should be explained what the location 4 is and that within the location 7 points from the IR camera were analysed. Same applies to Figs. 8-11. The authors could add sunset and sunrise to the figures.

Figure 15: The model configuration options should be explained in the figure text. Same applies to Fig. 16.

Figure 17: Please add y-axis to the plot a).

---

## Referee Comment (RC2) · Anonymous Referee #2 · 20 Apr 2017

General comments The authors present an addition to the well-known PALM Large Eddy Simulation (LES) model that addresses the effect of urban areas on local urban climate with a clear focus on the radiative temperatures of urban surfaces. In my opinion the approach that the authors seek is quite interesting and can be a valuable addition to detailed modeling efforts of radiative and turbulent transfer of heat and other quantities within the urban canopy. Unfortunately, the evaluation of the model against the available observations is quite poorly done or written down, and lacks a proper discussion of the strengths and weaknesses of the modeling approach. Furthermore, in my view a proper identification of mechanism that explain differences between modeled and observed temperatures is lacking. Also, I have the impression that the observational data set that the authors use to evaluate their model is too limited to fully appreciate the strength and weakness of the USM addition to the PALM model. Unfortunately,

my advice to the editor will thus be to reject the paper in its current form.

Major points Introduction • The section 1.2 lacks clarity in a number of places. Different processes with respect to urban climate are in my view not properly explained. • Also, section 1.2 of the introduction lacks a comprehensive review of current approaches to calculate the effects of urban climate. It mentioned different approaches, but it remains unclear to the reader how this fits into the method that the authors develop. Why did the authors for instance choose to develop their own urban scheme and not to adapt a current scheme for standalone models for use in PALM? Section 2 • The authors use the parameterization the PALM-LSM formulation for horizontal surfaces. Monin-Obukhov Similarity Theory (MOST) generally applies to the surface layer which bottom is typically about 3 times the height of the roughness elements. Could the authors justify why MOST is still applicable when grids are used with grid lengths that are in the order of few meters. • On page 6, from line 4 up to line 10, the authors outline a number of processes which they did not consider. I think that the authors should indicate why these simplifications are justified and what the impact is of these simplifications. For instance, I am not sure why the absorption of long wave radiation of the plant canopy can be neglected. Also, it is well known that the temperatures of plant canopies can differ considerably of the temperature of adjacent air layers. • In the line 5 on page 7, it is stated that the RCSF "represents the proportion of the radiative flux carried by the ray at its origin", while I interpret the first term within brackets on the right hand side of equation (8) as the effect of the attenuation f the original array by plant canopies in grid boxes that the ray crosses while going from A to C. Please clarify. • Page 9, line 11 to 13. It is not clear to me why the plant canopy has zero thermal capacity and why it should be applied to the grid box's air volume. In my view the absorbed radiation is used to heat up to plant canopy which on its turn exchanges sensible and latent heat with the surrounding air. Please, clarify • Section 2.4, page 10 line 4 to 11 form partly a repeating of the limitations already addressed in on page 6, line 5 to 10. Unfortunately, also here no justification has been provided. Section 3

• Page 10, line 14 to 28: no general description of the observation site is given. For instance the description lacks a reference to an exact location (lat/lon) or a general description of the morphological and urban structure characteristics at the observational site. Though there is some description of the sensors provided, important parameters such as view angle are lacking. • Page 11, page 5 to 13 It is not clear to me what the authors mean by "an indicative measurement of air temperature". Both the measurement technique and the location are poorly described and as such, too little detail is given to fully appreciate the values in this measurement in validating the USM in PALM. • Page 12, line 19: As the latent heat flux is usually small in urban areas, atmospheric boundary layers are usually deeper over urban areas than over the surrounding rural areas. Is it verified that the a height of 2364 m is high enough to ensure that the top of the ABL is below this value? • Page 13, line 6 to 13 I think that assigning appropriate values for model parameters is at least equally important that including the correct physical processes within a model. In my view this justifies a comprehensive description of the methods that have been used to calculate the input parameters, which is currently quite short. • Page 13/14, line 26 to 34. The line of argumentation by the authors is quite hard to follow. I agree with the authors that fig. 3 shows a lot of detail, but I do not see the lateral variations of simulated temperatures due to urban characteristics. I can see that the lower parts of the south facing facades are considerably cooler because of shadowing. Also, it is not clear to me how I can derive from the figure that local shading effects by subgrid sized faces are not captured by the model. • On page 15, line 1 to 8, the authors claim that the PALM USM does a good job at calculating the temperature evaluation at location 1 and in comparison with the Klementinum stations. This contrasts with the simulations with the WRF model which show a large bias. In my opinion, evaluating models on just one day is quite limited, and might lead to biased conclusions. Also, I am a bit surprised about the large bias in the WRF model. We know that it has a tendency to produce too low temperatures, but a difference between 12 C and 18 C is quite large. I am wondering whether it is an error in the model or whether the grid cell used to define the 2 m temperature has a

'rural land use' rather than an 'urban' land use. To gain more insight into this, it might be an idee to include a line in figure 6 giving the simulated and observed temperatures at an appropriate rural reference stations.  c Page 15, line 14 to 20: the line of argumentation of the authors. The authors claim that in figure 7, there are differences in observed temperatures because of the differences in insulation properties. At the same time, the authors claim that modelled temperatures at points 3 and 4 because of 'almost identical surface and material parameters' The model temperature line for point 4 in figure 7 is quite hard to distinguish, but it is clear to the temperature line at point 5 deviates from the model temperature line at point 3, but also from the temperature line at point 6 and 7. In contrast, for the other sites, differences in model nighttime temperature are remarkably similar.  c Page 17, line 4 to 9 and page 18, line 1 to 4: the authors identify some discrepancies between modelled and simulated temperatures, providing some speculation on which processes cause these discrepancies. I think that the paper would be much stronger when the authors include a sensitivity analysis substantiating the different processes that cause differences modeled and observed temperatures.

Conclusion  c In my view the conclusion section is much too general. At least the most important aspects of the model evaluation should be addressed and summarized.

Minor comments

 c Though parts of the paper are remarkably well-written (section 4), many other sections are quite hard to read, even for an experienced reader. I suggest that when the authors re-submit their paper, they go through their manuscript with a sharp pencil.  c Also, the manuscript contains many typographic errors and misspellings.  c A number of figures, for instance figures 7 to 11 hard to interpret.  c Maps, such as figures 4 and 5, 11, 12 and 13 lack lat/lon coordinates.

---

## Author Comment (AC4) · 1 Aug 2017

Please find attached manuscript changed based on the reviewers comments together with supplements.

Please also note the supplement to this comment:
https://www.geosci-model-dev-discuss.net/gmd-2017-61/gmd-2017-61-AC4-supplement.zip

---

## Author Response (AR1)

A. Kerkweg

kerkweg@uni-bonn.de

Dear authors,
in my role as Executive editor of GMD, I would like to bring to your attention our Editorial version 1.1:
http://www.geosci-model-dev.net/8/3487/2015/gmd-8-3487-2015.html
This highlights some requirements of papers published in GMD, which is also available on the GMD website in the 'Manuscript Types' section:
http://www.geoscientific-model-development.net/submission/manuscript_types.html
In particular, please note that for your paper, the following requirements have not been met in the Discussions paper:

• *"The main paper must give the model name and version number (or other unique identifier) in the title."*
*In order to simplify reference to your developments, please add a version number of the USM in the title of your article in your revised submission to GMD. For easier recognition by the reader it might also be helpful to add the acronym to the title.*

The version name (PALM-USM v1.0) was added to the title.

Yours,
Astrid Kerkweg
*The manuscript A new urban surface model in the large-eddy simulation model PALM by Resler et al. describes the addition of new radiation and energy model (USM) to the existing large eddy simulation model PALM. This is an important addition to PALM, which before did not account for radiation transfer in complex street canyons well. This is important for several urban processes and thus the new model clearly is a welcomed addition to the capabilities of PALM. There are however some challenges, what comes to representativeness of the new module and presentation of the methods and results (see below), in the manuscript that needs to be addressed. Also the language of the manuscript needs further improvements. There are several parts that need revision and in the minor comments I've tried to point out some of them, but I suggest the authors to go through the language once again before resubmitting the manuscript. Thus I suggest major revisions to the paper before it can be accepted for publication to Geoscientific Model Development.*

We would like to thank the reviewer for the evaluation of our manuscript and useful comments that considerably helped to improve the manuscript. Both reviewers raised important points that led to an extensive revision and also to the need to re-run the presented simulations which are therefore different from the previously submitted version. We have addressed all comments and our responses are stated below. Reviewer comments are in italics and authors responses are in blue standard font. The revised manuscript with marked-up changes is attached at the end of this document. We added the references to the sections or pages/lines into the responses to ease location of changes made wherever relevant. Together with this response we uploaded a revised manuscript and supplements.

Major comments

Representativeness of the new module

*To my understanding the anthropogenic heat emissions only from traffic are accounted for. This might not be an issue in summer at the site of evaluation as there are no heat emissions from buildings, but what if the model evaluation would have been made in winter*

*or in other city with huge need for air conditioning? This is a clear lack in the USM as the authors could have implemented e.g. a simple temperature related anthropogenic heat emission model following some activity profile similar to the traffic heat emissions.*

The current version of the model allows to prescribe any anthropogenic heat with fixed diurnal profile of its release while it has no meteorology-dependent anthropogenic heat model. The limitation to heat emission from transportation is the design of the presented study, which is justified by the character of modelled area and season. The corresponding formulations were supplemented to the text and present text was clarified. The new section 2.3 was added and the description of settings of anthropogenic heat in the study (section 3.2.5) was extended including the reference to our previous sensitivity study of the influence to heat from transportation. The discussion of limiting the anthropogenic heat to heat from transportation was added to the new Sect. 4 Discussion (P33, L6-8). The section Conclusions was extended by information about preparation of the coupled building energy model (work in progress) (P36, L31-33).

*Also, USM neglects latent heat flux component from the surface energy balance, which can be important in neighbourhoods with more vegetation. At the same time I understand why in the first step of the model development only the some (most crucial?) points of the complex energy system are included, but the authors should still comment in more detail about the limitations of the model. Some limitation is currently described on P10, but the representativeness and limitations of current USM model version should be described in detail either in the results/conclusions or in a separate section after the results.*

The reviewer is right, the latent heat flux is very important part of the energy balance in many cases and it was not sufficiently stated in the submitted text. To better cope with this topic, we clarified the intention of the first version of the model to simulate UHI situations which leads to selection of the implemented processes in the first version. We also added new Sect. 4 with discussion of the current limitations including a paragraph which justifies this omission in the conditions of our modelled situation including a reference to the literature (P31-32, L28-3). We also extended the conclusions by information about the prepared extensions to the USM which will include also the treatment of the latent heat.

*On P6, L3-10, the authors list radiation-related processes that were omitted from the radiation model. Could the authors add what is the level of impact these omitted processes might have on the model performance?*

The list of omitted radiation-related processes has been moved to Section 4 (Discussion) and extended with descriptions of impact.

*Surface properties in USM*
*Please add somewhere to Section 2.3 that the needed surface properties to run USM are given also in the Supplementary material.*

Statement that all needed surface and material parameters are listed and described in supplements was added to Sect. 2.4 (P10, L29-30). Two tables (Table S2 and Table S3) showing parameters used in the presented evaluation case study were also added to supplements.

*How is the clearness index (P5, L26) given to the model?*

Clearness index is a standard measure of atmospheric attenuation of solar radiation. As defined within the referenced article, it is the ratio of global horizontal irradiance (GHI, at ground level) to extraterrestrial solar irradiance (ETR, at top of atmosphere). GHI is known directly from the PALM model and ETR is essentially the solar constant adapted for orbital eccentricity and multiplied by cosine of solar zenith angle. We feel that these details are only relevant when studying the method in the referenced article, therefore the mention of clearness index has been removed.

*For the evaluation part, it would be important to know what surface property values were used for the different observational points presented in Figs. 7-11. Maybe a table to the main paper or supplementary material would work? Also in the results the effect of the different properties could be extended.*

We added tables with material properties for all evaluation points to supplements (Tables S2 and S3). We also extended result section with sensitivity analysis on material parameter values (Sect. 3.4.2).

*What was the anthropogenic profile you used for traffic emission? It would be good to plot this together with traffic rates and meteorology for the case study period (see comment below). The obtained traffic heat emissions (P13, L19-20) are rather large during peak traffic hours. To me they seem unrealistic so could the authors comment how they compare with other studies.*

We extended the paragraph about traffic heat rates (Sect. 3.2.5) with more detailed description of traffic heat calculation. We added a diurnal profile of average traffic heat flux to a new figure with meteorological variables (Fig. 4). The presented values referred to heat fluxes right in the traffic lanes while it is common to present the anthropogenic heat rates averaged over larger area - e.g. whole city or over a grid cell of size of hundreds of $m^2$. We added this clarification to the text together with averaged value over the whole model domain to ease a comparison with other studies. The estimates of anthropogenic heat differs considerable depending on the used methodology. Sailor (2011) reviewed different methods for estimating anthropogenic heat in the urban environment. The estimates for average (total) anthropogenic heat ranged from 9 - 150 W $m^2$, summer values being considerably lower than winter values. Our estimate of 2 W $m^2$ seems to be reasonable as we are considering only traffic heat in the area with only moderate traffic.

*Model runs*
*The vertical domain height is high when compared to the horizontal scale of the simulated area. At the same time the authors say that outside domain area has minor impact on the processes within the modelling domain, but such a high vertical domain makes me doubt this. This must be affected by some further away surface not with similar characteristics as the study area. Could the authors comment this?*

The reviewer is right. The horizontal model domain is too small, which basically imposes some limitations in resolving the largest turbulent eddies (comment added, P14-15, L12-2). Ideally, these would arrange themselves into hexagonal cellular patterns that scale with the height of the boundary layer. In our case, this height was about 2000 m during daytime. In this context, the horizontal model domain was way too small. However, this will not affect the available energy in the system. Our recent experience is that the feedback between turbulent eddies and bare soil (which behaves similar to solid walls) is rather small - meaning that incorrect representation does not have to lead to major drawbacks. Nevertheless, the vertical profiles of potential temperature (see new Fig. 10) as simulated display untypical unstable stratification during daytime, which appears unrealistic. The test run included in the revised version (see Fig. 24) with a larger horizontal domain clearly shows that this can be removed by extending the model domain which allows a free development of turbulence. The temperature profile then was nearly-neutral as expected under convective conditions. For our results, this limitation leads to too high air temperatures within the canopy, which potentially affects the interaction with the surfaces. We discuss this effect in the revised manuscript (P32, L5-29). Unfortunately, we did not have sufficient computational resources to perform a run with a sufficiently large domain (say about 5 x 5 km). Our experience showed that the current version of the USM, particularly the calculation of the sky view factors does not scale well with increasing grid points which leads to a memory problems.

We are currently working on a solution for this issue to make the application of the USM feasible on large domains.

The reviewer is also questioning whether there might be significant impacts from outside the analysis domain. This is of course very well possible. However, under very low-wind conditions (as in the present study), very local (street-canyon-size) effects dominate the local processes and larger-scale impacts (from regions far away) are of minor importance. Given the good agreement between simulation results and measurement data, we are confident, that the limitation of the horizontal model domain poses no major limitation to this first evaluation. In the future, we plan to perform larger-scale setups were we will perform sensitivity tests regarding domain size so that this reasoning will be put on a more solid foundation.

We also changed the statement about minor impact of the outside domain. The original formulation was inaccurate. The current formulation better corresponds to authors original intention (P15, L4-6).

*It is not explained clearly why did the authors use WRF data to provide forcing for the run. This is shortly described in the results (P15, L4-5, 7-8) but the explanation should be given already in section Model setup. How did the model forcing data look relative to the Karlow station data? Air temperature data is given in Figure 8, but how did wind look like? I suggest that new figure where meteorological variables from WRF and observations (Tair, wind, solar radiation) and traffic rates for the simulated period would be plotted.*

To account for the processes occurring on larger scales than modelling domain, but still affecting the processes inside the domain, we employed the large-scale forcing and nudging option of PALM. As no observation data are available in needed vertical structure and time resolution, we used the WRF data to obtain needed forcing values. We extended the large-scale forcing description (section 3.2.3) by this explanation (P15, L16-19). We also added three figures showing the comparison of WRF data against both ground (Figs. 6 and 7) and sounding (Fig. 8) measurements. Other meteorological values from Karlov station together with traffic rates are plotted in the new Fig. 4.

*Results*
*The model evaluation section is currently quite poorly written and needs revisions. Text on P15, L13-19 is unclearly written and jumps between differences in the observation points, comparison between model output and observations and furthermore locations. Also, is this part referring only to location 1 as its not clear from the text. If yes then the general*

*conclusion that modelled wall temperature drops faster after sunset is not valid as only on half of the points this is the case and in half not. Rather this pace of cooling could be related to thermal properties of the different points. Please rewrite.*

Model evaluation section was completely rewritten and changed according to the new model runs. We tried to make it easy-to-read and moved discussion into the relevant new section.

*On P17, L4, the overestimation takes place only in daytime. Please add this information. The authors mention here that the daytime overestimation could be due to heat capacity of the wall. This could be the case indeed as it seems that the surface is not storing enough energy in daytime and release it enough in night-time. This should be discussed more properly in the results section.*

All text was completely reformulated and the relevant discussion was added to Discussion section.

*In generally more text about the surface properties and their impact to the model performance should have been added. I'm missing some sensitivity tests about the impact of surface properties to the model performance. For example the authors could choose some location point from Figs. 10 and 11 where the surface properties would be slightly changed and improve relative to the observations. This particularly in the case of Figure 11, where the surface temperatures seem to be completely off.*

We added whole new section 3.4.2 - Sensitivity to material parameters. The sensitivities of the surface temperatures to albedo, roughness and thermal conductivity decrease or increase are described in this section.

*The authors could add more analysis on section 3.5 about the differences between PALM without and with USM. How great impact does the addition of USM have on the turbulent mixing. Could maybe some spatial means at different heights be calculated to really see how mixing is improved? Or maybe showing vertical profiles from certain points on the main streets? Due to missing measurements, I guess the authors cannot really comment is the representation of turbulent mixing improved or not.*

We performed a new sensitivity study where we compare the basic run of PALM-USM model with an run with USM module switched off and fixed heat fluxes prescribed for all surfaces. We compare the resulting flow in the street canyon in the new section 3.4.1. The reviewer is

right that we are not able to compare the results to any measurements and to prove the improvement of the representation of turbulent mixing now.

*The problems related to model/observation comparisons are not mentioned in the conclusions. Possible needs to improvements should be added there.*

We added some discussion about the measurements uncertainty to the Discussion section (P33, L9-12).

*Minor comments*

All the minor comments which concern some particular formulation were carefully incorporated into the text. Since the text went through a deep revision, many affected parts have gone of it. We thus write the specific answer only to the comments which are relevant in the new text.

*P1, L1: "a direct effect" -> "direct effects"*

accepted

*P1, L2: "This implies the need for a reliable tool for climatology studies that supports urban planning and development strategies" -> "This implies that reliable tools for local urban climate studies supporting sustainable urban planning are needed"*

accepted

*P1, L4-5: ". . .a new Urban Surface Model (USM) describing the surface energy processes for urban environments was developed. . ."*

accepted

*P1, L7: In the model the authors neglect latent heat flux and thus are not calculating the total energy balance for impervious surfaces. Please reword here.*

This omission is mentioned in the description of the surface energy balance equation and discussed on the Sect. 4.1.

*P1, L9: Please open what MPI means.*

We added "the standard Message Passing Interface (MPI)" (P1, L11) and the reference to the official web pages (P10, L24)

*P1, L19: I would remove the first sentence: it is said in the abstract already.*

accepted

*P1, L20: Add "in future" after increasing, add "local urban" in front of climate.*

accepted

*P2, L2-3: I would reword this difficult sentence as e.g. it is not clear what is meant with "sound scientific background". I guess the authors mean rather tools?*

The sentence was reformulated.

*P2, L4: Should be ". . .phenomenon related to. . .". I would change the UHI reference to the original paper by Oke.*

The sentence has been changed to "One major phenomenon related to the urban climate..." and the reference updated to the original paper Oke 1982.

*P2, L6: ". . .retention energy of urban surfaces and increased heat emissions from human activities."*

accepted

*P2, L7: Not only building shadows create cool islands but also tree shadows and increased evaporation.*

The entire paragraph was rewritten.

*P2, L7-L27: In these lines there is unnecessary repetition and should be restricted. After the cool island should be the whole description how the heat islands are commonly studied and after that what problems these methods meet so that eventually LES modelling is required to understand the issue. Also on some lines the authors talk about urban processes generally and on some lines only on the urban heat island. Also the references on L17 consider only UHI and not e.g. air quality that the authors mention on L2 at the same page.*

This part of the text has been rewritten.

*P2, L33: can be -> is*

accepted

*P3, L3: Remove comma from the front of LES.*

accepted

*P3, L3-4: "Many of the CFD models do not contain appropriate radiative models and to overcome this deficiency, an independent radiative models with the resulting radiation fluxes have been imported into the CFD model. . ."*

The paragraph has been rewritten.

*P3, L9-16: The objectives of the manuscript focus now on the project under which the project is made of, but these should be rephrased to be more general and representative for the actual study.*

The reference to the project was removed and the part rewritten in the more general way.

*At the same time LES does not require CFD in its front so please remove it.*

Accepted, removed.

*P3, L18: Abbreviation PALM should be opened in the text here.*

The authors of the PALM decided to drop the treatment of the name as an abbreviation and they consider it just a name now. (It is not done on the PALM web page yet.) We respect this movement and do not include the older long name in our text.

*P3, L28: ". . .obstacles as well as the landform"*

accepted, added to the sentence

*P3, L31: Replace next with Secondly; ". . .radiative exchange at the surface. . ."*

accepted

*P4, L2: of using -> to use*

accepted

*P4, L7: ". . .PALM-LES, further extends the surface parameterisations. . ."*

accepted

*P4, L12 ". . .plant canopies have not been. . ."*

accepted

*P4, L16: Again only radiation and direct heat flux is considered: not the whole energy balance*

A new paragraph, which briefly describes the limitation of the current version, was added to this part of the text. These limitation are consequently discussed in the following text, mainly in Sect. 4.

*P4, L19: as well as -> and; material -> materials*

accepted

*P4, L24: heat fluxes -> sensible and storage heat fluxes. Also I would add already here at the end of the sentence that heat consumed to evaporation is not accounted for.*

Clarification added

*P4, L25: "The energy budget in the skin layer. . .". The reference to PALM-LSM is not needed here again.*

accepted, the reference to LSM paper removed

*P4, L26: Anthropogenic heat flux is missing from the equation.*

Anthropogenic heat is not considered in the surface energy balance equation intentionally. As our primary intention was to add the anthropogenic heat from transportation, we model the release of the heat directly into the air. We added a new section 2.3 Anthropogenic heat which describes the implementation of the anthropogenic heat in the model.

*Units are missing from the variable descriptions (and also from later equations). Please add throughout the manuscript.*

Rather than adding the units to all variable descriptions throughout the manuscript we opted to create a comprehensive table where all variables are listed together with units and its description. This table can be found in supplements as Table S1, reference was added to the text (P5, L25-24).

*P5, L3: Why here the potential temperature is used whereas in Equation (1) there is air temperature? Shouldn't zero refer to skin surface and not surface?*

The potential temperature is used in there parameterization of H as it is the prognostic quantity in PALM and also because the use of Monin-Obukhov Similarity theory requires the use of potential temperature to account for the correct buoyancy. In the parameterization of G, however, the actual temperature must be used as there is no potential temperature within the solid material. Actually, in the code, potential temperature is converted into actual temperature using the Exner function in order to solve for the skin surface temperature.

*P5, L4-7: Could the authors add a bit more information about the parameterizations especially as the Maronga and Bosveld paper has only been submitted.*

The paper of Maronga & Bosveld is currently in press and available as online version already. It gives an outline of the land surface scheme. As most parts follow the ECMWF scheme, we added a citation to it as well. For the interested reader, the full description is given on the PALM homepage. A manuscript which is purely dedicated to the land surface scheme is currently under preparation.

*P5, L13: Should be systematically PALM-LSM.*

accepted

*P5, L16: You can replace "Ground heat flux" with "G".*

replaced by "The flux G"

*P5, L17: Following equation 1 the layer next to surface should be skin layer?*

accepted, reformulated

*P5, L22: The title could be "Multi-reflection transfer model" as then it would be systematic with the text on P4, L20-22.*

changed to "Radiative transfer model" to be consistent within the text

*P5, L29-L30: The processes related to shortwave radiation is unclear. It is written that process "Radiation sources from the sun. . .using the relative position of the sun" is modelled, but from the above text I get the impression that the shortwave radiation on top of the canopy is obtained from the chosen radiation module in PALM. Thus, please be more specific here.*

The section 2.2 was clarified and partly rewritten.

*P6, L12: Here the authors use word irradiance at each surface whereas in the energy balance equation (1) they use net radiation. Please, systematize throughout the manuscript.*

Net radiation is the total radiative budget (i.e. incoming irradiance minus reflected radiosity and emitted radiosity). In this paragraph we wanted to emphasize that following methods are used to model the irradiance, unlike the outgoing radiation which is readily available, therefore we have kept the distinction in the text.

*P6, L16: Remove "also" from the sentence*

accepted

*P6, L20: The abbreviation for the differential view factor (uppercase d should be give here).*

accepted, added

*P6, L21: As this is generally the equation used for sky-view factor I would add a reference to the equation.*

Reference added

*P6, L23: The separation distance is explained in the previous sentence on the same line and thus abbreviation s can be used after "Under the assumption. . ."*

accepted

*P6, L26: Please explain what A' means. In generally text and equations are not very clear starting from here and ending on P7, L2 and additional information source needs to sought if you are not that familiar with the calculation of view factors. Thus I suggest the authors to add a bit more explanation to this part of the manuscript with proper description of the variables used in the equations.*

A' is the iterator for the sum of all view factors having the same target face. The text has been revised.

*P6, L8: Same applies to Equation (5). It is not explained that this equation is valid for the case where two canopy grid boxes C and D are between surfaces A and B. First it should be given what is the RCSF for a single grid box C or D.*

The text has been thoroughly revised. The view factor geometry calculations are all done before accounting for plant canopy and the attenuation by plant canopy is applied separately, as stated near the end of Section 2.2.2.

*P10, L16-28: The order of explanation is strange here. The authors first describe the measurement locations before explaining what instruments are used. I would suggest to explain first what is measured and how (surface temperature using infrared camera) and then the actual locations of the measurements. How far was the camera from the surfaces and what was its view in degrees.*

The whole Sect. 3.1.1 Measurements was reordered and explanations extended.

*P11, L22: What is meant with "slight changes in camera position"?*

The explanation was added in the text (P13, L4-5). In the original version we wrote "...to correct for slight changes in camera position during measurement". Preposition "during" might have been confusing. Slight changes in camera position were result of the fact that camera was carried from one location to another each hour.

*P12, L4: It would be better to describe here the selected surface cover types and not in the results section.*

The text about selection of surface cover types (evaluation) points was added (P13, L5-13).

*P12, L5-6: How was air temperature measured?*

We added the description of temperature measurement device and its uncertainties (P13, L14-18).

*P12, L7-12: It would be nice to have the meteorological conditions plotted in a Figure from the around 1.5 day measurements campaign (see major comments)*

We included a new figure with meteorological conditions as Fig. 4.

*P12, L29: What is Medard prediction system?*

We removed the confusing Medard name, which has no added value to a reader.

*P13, L24-25: It would be nice to have these times in the meteorological figure as lines or as radiation itself.*

The nighttime and noon was added to relevant figures.

*P13, L26: Add "modelled surface temperatures"*

accepted

*P13, L27: It is better not to use the name of the street when referring but rather use "along the west-east street"*

accepted

*P14, L4: ". . .of the domain to illustrate the effects of tree. . ."*

Reformulated as the figure was moved into supplements.

*P15, L1: I would remind here what kind of measurement location location 1 was.*

Text was rewritten and location better specified.

*P15, L2: What is meant with indicative measurement? The automatic weather station is not mentioned in methods and thus should be added there.*

We added a description of all used meteorological stations in section 3.1.1. The meaning of term "indicative measurement" was explained (P13, L14-18).

*P15, L3: Klementinum complex does not say much to the reader. Is this large area? Is the station part of official meteorological monitoring? This should all be added to the methods.*

We added a summary description of all weather stations in section 3.1.1 and changed the description of Klementinum.

*P15, L5: I would simplify the sentence: "The street level air temperature form PALM-USM is in. . ."*

accepted

*P15, L7: ". . .temperatures. . ."*

accepted

*P15, L9-10: "Comparisons. . .are displayed. . ."*

accepted

*P15, L10: ". . .observed temperature patterns. . ."*

accepted

*P15, L26 onwards; I would move Figure 5 to Supplementary material as there are already many figures, and new should be added. The figure is nice looking but not relevant for the actual paper.*

The figure was moved to supplements.

*P18, L2: plays -> play. Please open here what do you mean by these effects.*

Whole section was reformulated.

*P22, L1: had been -> was*

accepted

*P22, L1: ". . .in the range. . ."*

accepted

*P23, L7-8: only sensible heat flux is given.*

The section Conclusions was completely rewritten.

*Figure 2: Add scale also to this aerial image similarly to Fig. 3. You could also draw the area of the observational image to Fig. 3 in a similar fashion as you show the modelling area with green.*

Scale was added to the figure (current Fig. 3).

*Figure 4: Figure text needs more explanation: It would be good to add the date and that the data shown is modelled.*

The description was enhanced.

*Figure 5: Scale is missing from the figure. Would it be possible to add this small area to Fig. 3 as a box?*

As the figure was moved to supplements we decided to leave out the box in Fig. 3. We added lon/lat coordinates to figure description instead (Fig. S11).

*Figure 6: It is difficult to see the lines if printed in black and white. Some of the darker colour lines could be plotted as dashed lines to separate them. I think AGL is not explained in the manuscript. Also as the figure should be able to be looked without references to the text, the authors should add the location 1 above road to the figure text as well as the time (2-3 July 14:00-17:00).*

Figure was replotted to allow for BW print. The date was added and so was the reference to location 1 in figure caption. Explanation of AGL was added to the manuscript (P26, L13-14).

*Figures 7-11: The figure texts are very poor currently. Please modify them to include the day, time period, and on the first one also description about the solid and dotted lines. Also in Figure 7 it should be explained what the location 4 is and that within the location 7 points from the IR camera were analysed. Same applies to Figs. 8-11. The authors could add sunset and sunrise to the figures.*

Figures were modified, time of sunset, sunrise and noon was added.

*Figure 15: The model configuration options should be explained in the figure text. Same applies to Fig. 16.*

The sentence "The setup of the model corresponds to the setup described in with reduced number of layers to 81." into the description of Fig. 15 (current Fig. 25). The term "Model configuration and…" was added to the description of Fig. 16 (current Fig. 26).

*Figure 17: Please add y-axis to the plot a).*

The description "Effectiveness of parallelization" was added to y-axis.

**References**

Sailor, D. J.: A review of methods for estimating anthropogenic heat and moisture emissions in the urban environment, *Int. J. Climatol.*, 31(2), 189–199, 2011.
*The authors present an addition to the well-known PALM Large Eddy Simulation (LES) model that addresses the effect of urban areas on local urban climate with a clear focus on the radiative temperatures of urban surfaces. In my opinion the approach that the authors seek is quite interesting and can be a valuable addition to detailed modeling efforts of radiative and turbulent transfer of heat and other quantities within the urban canopy. Unfortunately, the evaluation of the model against the available observations is quite poorly done or written down, and lacks a proper discussion of the strengths and weaknesses of the modeling approach. Furthermore, in my view a proper identification of mechanism that explain differences between modeled and observed temperatures is lacking. Also, I have the impression that the observational data set that the authors use to evaluate their model is too limited to fully appreciate the strength and weakness of the USM addition to the PALM model. Unfortunately, my advice to the editor will thus be to reject the paper in its current form.*

We would like to thank the reviewer for the evaluation of our manuscript and useful comments that considerably helped to improve the manuscript. Both reviewers raised important points that led to an extensive revision and also to the need to re-run the presented simulations which are therefore different from the previously submitted version. We have addressed all comments and our responses are stated below. Reviewer comments are in italics and authors responses are in blue standard font. The revised manuscript with marked-up changes is attached at the end of this document. We added the references to the sections or pages/lines into the responses to ease location of changes made wherever relevant. Together with this response we uploaded a revised manuscript and supplements.

Major points

Introduction

*The section 1.2 lacks clarity in a number of places. Different processes with respect to urban climate are in my view not properly explained. Also, section 1.2 of the introduction lacks a comprehensive review of current approaches to calculate the effects of urban climate. It mentioned different approaches, but it remains unclear to the reader how this fits into the method that the authors develop. Why did the authors for instance choose to develop their*

*own urban scheme and not to adapt a current scheme for standalone models for use in PALM?*

The introduction was reworked to be more clear and comprehensive. We also tried to briefly indicate the reasons which lead as to the decision to develop a new model of urban canopy for PALM (P3-4, L32-3). As far as current approaches are concerned, we only mention the processes and techniques that are relevant to the present study and we are aware that this does not constitute a thorough review. However, thanks to the fact that a number of recent comprehensive reviews is available (e.g. Mirzaei&Haghighat, 2010 and Mirzaei, 2015, that we cite), therefore we wouldn't like to substitute them and we would rather concentrate on description of PALM-USM.

Section 2

*The authors use the parameterization the PALM-LSM formulation for horizontal surfaces. Monin-Obukhov Similarity Theory (MOST) generally applies to the surface layer which bottom is typically about 3 times the height of the roughness elements. Could the authors justify why MOST is still applicable when grids are used with grid lengths that are in the order of few meters.*

Traditionally, MOST is applied over flat horizontal surfaces. For non-building resolving simulations, the grid spacing is coarse so that the requirement that the first grid point is within the surface layer (where MOST holds) is usually fulfilled. In building-resolving simulations, where we can resolve the buildings themselves by a large number of grid points and where we have to deal with both horizontal and vertical surface elements, we need an approach to describe the turbulent exchange of all these surface elements with the atmosphere. There is no theoretical framework available for this application, especially for vertical walls. Here, the use of a stability function is questionable. The established and widely-spread approach is to apply MOST locally for all surface elements (horizontal and vertical). Previous attempts to do this with PALM have been successfully validated e.g. against wind tunnel measurements (Letzel et al. 2008, Kanda et al. 2013, Park & Baik 2013). MOST is used here, despite it is known that some assumptions of the theory are violated. There is no physical justification for this, but the results indicate that this approach gives very reliable results. MOST is hence used in all building-resolving microscale models and is the basis of surface schemes such as SUEWS and TUF-3D. We added a more rigorous justification for using MOST and the relevant citations to the mentioned validation papers to the manuscript (P6, L1-5).

*On page 6, from line 4 up to line 10, the authors outline a number of processes which they did not consider. I think that the authors should indicate why these simplifications are*

*justified and what the impact is of these simplifications. For instance, I am not sure why the absorption of longwave radiation of the plant canopy can be neglected. Also, it is well known that the temperatures of plant canopies can differ considerably of the temperature of adjacent air layers.*

The description of the omitted processes was completely rewritten. We also mention the purpose of this first version to model UHI situations during the summer heat wave episodes which affected the choice of the implemented processes in this version. We added a new Sect. 4 Discussion where we discuss the limitations of the model and the setup of experiment and we try to assess the potential impact to our results. Finally, we complemented the conclusion with the short description of the current and future development of the PALM-USM.

*In the line 5 on page 7, it is stated that the RCSF "represents the proportion of the radiative flux carried by the ray at its origin", while I interpret the first term within brackets on the right hand side of equation (8) as the effect of the attenuation of the original array by plant canopies in grid boxes that the ray crosses while going from A to C. Please clarify.*

The attenuation by grid box C has to be multiplied by the complement of sum of attenuations before reaching C on the path from A. This way we get the proportion against the original radiative flux and not just against the radiative flux that reaches box C. The text has been revised for clarification (P8, L4-5).

*Page 9, line 11 to 13. It is not clear to me why the plant canopy has zero thermal capacity and why it should be applied to the grid box's air volume. In my view the absorbed radiation is used to heat up to plant canopy which on its turn exchanges sensible and latent heat with the surrounding air. Please, clarify*

For our level of detail of plant canopy modelling, we decided to use a simplification where the absorbed heat flux is exchanged with the air instantaneously, just like it is done in some other radiative transfer models and in accordance with the current implementation of non-urban plant canopy model in PALM. Discussion of this simplification has been added to Sect. 4 (P31, L24-27).

*Section 2.4, page 10 line 4 to 11 form partly a repeating of the limitations already addressed in on page 6, line 5 to 10. Unfortunately, also here no justification has been provided.*

We reorganized the text about limitations to avoid repeating. We state the general limitations at the beginning of the Sect. 2 (P5, L13-16) and radiation related processes in the Sect. 2.2.1 (P7, L5-7). We added the whole discussion section (Sect. 4), where applicability and the possible impact on the presented results of these limitations is discussed.

*Section 3*
*Page 10, line 14 to 28: no general description of the observation site is given. For instance the description lacks a reference to an exact location (lat/lon) or a general description of the morphological and urban structure characteristics at the observational site. Though there is some description of the sensors provided, important parameters such as view angle are lacking.*

Whole Sect. 3.1 was extended and reorganized. All locations are given lat/lon coordinates, general description of the morphological and urban structure characteristics was added. Description of sensors was added/extended.

*Page 11, page 5 to 13 It is not clear to me what the authors mean by "an indicative measurement of air temperature". Both the measurement technique and the location are poorly described and as such, too little detail is given to fully appreciate the values in this measurement in validating the USM in PALM.*

The whole Sect. 3.1 about observation location and measurements techniques was rewritten and extended to better describe the campaign. An explanation of what is meant by "indicative measurement" was added (P13, L14-18).

*Page 12, line 19: As the latent heat flux is usually small in urban areas, atmospheric boundary layers are usually deeper over urban areas than over the surrounding rural areas. Is it verified that the a height of 2364 m is high enough to ensure that the top of the ABL is below this value?*

The reviewer is right. The domain height was too low, indeed. In the course of the revision we also revised the model domain height which is now about 3.5 km, while the boundary layer during daytime is about 2 km, so that the new domain height can be considered to be high enough. See also the vertical profile shown in Fig. 5. The results indicate that the boundary layer in the city centre (as simulated by the USM) is indeed higher than what was measured using the radiosonde in suburban area. In that sense, the results appear to reproduce the expected behavior (see Figs. 10).

*Page 13, line 6 to 13 I think that assigning appropriate values for model parameters is at least equally important that including the correct physical processes within a model. In my view this justifies a comprehensive description of the methods that have been used to calculate the input parameters, which is currently quite short.*

The section 3.2.4 about input surface and material parameters was extended. We also added two tables to supplements (Table S2 and S3) with parameter values that were used in our case study.

*Page 13/14, line 26 to 34. The line of argumentation by the authors is quite hard to follow. I agree with the authors that fig. 3 shows a lot of detail, but I do not see the lateral variations of simulated temperatures due to urban characteristics. I can see that the lower parts of the south facing facades are considerably cooler because of shadowing. Also, it is not clear to me how I can derive from the figure that local shading effects by subgrid sized faces are not captured by the model.*

We consider this view as an illustrative one. We added a view to the east facing wall in two morning hours (Fig. 18) which better illustrates the effect of material properties. It is true that the local shading effects by subgrid sized faces cannot be directly observed from the figure (current number 17). We moved the discussion elsewhere in the text (P24, L10-14). The description of the figure was rewritten accordingly.

*On page 15, line 1 to 8, the authors claim that the PALM USM does a good job at calculating the temperature evaluation at location 1 and in comparison with the Klementinum stations. This contrasts with the simulations with the WRF model which show a large bias. In my opinion, evaluating models on just one day is quite limited, and might lead to biased conclusions. Also, I am a bit surprised about the large bias in the WRF model. We know that it has a tendency to produce too low temperatures, but a difference between 12 C and 18 C is quite large. I am wondering whether it is an error in the model or whether the grid cell used to define the 2 m temperature has a 'rural land use' rather than an 'urban' land use. To gain more insight into this, it might be an idee to include a line in figure 6 giving the simulated and observed temperatures at an appropriate rural reference stations.*

The urban parameterisation in WRF was intentionally not enabled in order to avoid double counting of the urban canopy effect which is treated by the PALM-USM model. This comment was added to the section 3.2.3 about large-scale forcing (P16, L6-7). WRF values in original Fig. 6 were 3-8 degC lower than presented measurements. However, WRF values are not directly comparable to displayed indicative and Prague, Klementinum measurements

as WRF should represent background effect in the WRF-PALM-USM system, while both measurements correspond to temperatures inside urban canopy. Thus we opted to omit the WRF from Fig. 6 (Fig. 9 in the current version) and based also on the suggestion of the other reviewer, we extended the section 3.2.3 with WRF comparison to relevant ground and sounding weather stations. WRF show bias of roughly 1-2 degC . Despite the slight cold bias of the WRF simulation, we take the WRF-derived values as the best inputs available.

*Page 15, line 14 to 20: the line of argumentation of the authors. The authors claim that in figure 7, there are differences in observed temperatures because of the differences in insulation properties. At the same time, the authors claim that modelled temperatures at points 3 and 4 because of 'almost identical surface and material parameters' The model temperature line for point 4 in figure 7 is quite hard to distinguish, but it is clear to the temperature line at point 5 deviates from the model temperature line at point 3, but also from the temperature line at point 6 and 7. In contrast, for the other sites, differences in model nighttime temperature are remarkably similar.*

Whole Sect. 3.3 was completely rewritten. The description of the comparison of modelled and observed values was significantly extended and clarified. We also added new Sect. 4 Discussion.

*Page 17, line 4 to 9 and page 18, line 1 to 4: the authors identify some discrepancies between modelled and simulated temperatures, providing some speculation on which processes cause these discrepancies. I think that the paper would be much stronger when the authors include a sensitivity analysis substantiating the different processes that cause differences modeled and observed temperatures.*

We performed tests of sensitivities to model setup parameters which could give some insight into one of the possible sources of differences. We also computed two idealized simulation. The first one illustrates the influence of the USM to the air flow in the street canyon and the other assesses the possible influence of the domain extent to the results. The sensitivity test are summarised in Sect. 3.4. We moved the discussion of the results and performed tests into Sect. 4 where also other issues of the model and setup are discussed.

Conclusion
*In my view the conclusion section is much too general. At least the most important aspects of the model evaluation should be addressed and summarized.*

We rewrote the section Conclusions to better describe the achievements and to reflect the new and enhanced results. We also complemented the short summary of the limitations of the model as well as the short description of the current and future development of the model.

Minor comments

*Though parts of the paper are remarkably well-written (section 4), many other sections are quite hard to read, even for an experienced reader. I suggest that when the authors re-submit their paper, they go through their manuscript with a sharp pencil.*

The manuscript is considerably rewritten, we believe that the text is more comprehensible now.

*Also, the manuscript contains many typographic errors and misspellings.*

The manuscript went through language corrections.

*A number of figures, for instance figures 7 to 11 hard to interpret.*

We increased the plot size together with the font size and reorganized the colour scheme. To make the plots easier to interpret we added some details (shading of the night time and solar noon line). We also extended the figure descriptions (current Figs. 12-16).

*Maps, such as figures 4 and 5, 11, 12 and 13 lack lat/lon coordinates.*

Coordinates of the crossroads, which is in the middle of the case study area, was added to the text. Observation locations are shown in Fig. 3 which contains url to web map, where these locations are displayed. We also added coordinates too all graphs with model/observation surface temperatures comparison.

**References**

[revised manuscript text omitted]

The output of the WRF model was compared to measurements from the four Prague stations (see Sect. 3.1.1). The overall
10  agreement between the simulated values and the observations is reasonable and corresponds with long term evaluations done earlier. For the period of 1–5 July (see Fig. 6), WRF shows a cold bias. The largest bias occurs in the urban Prague, Klementinum station (city centre), which is as expected given the urban parameterization not being enabled in the WRF model. On the other hand, the comparison with Prague, Kbely station (closest background station to the area of interest) shows only a small bias (see also timeseries in Fig. 7). Also the comparison with vertical profiles of temperature from Prague, Libuš
15  station shows good agreement (see Fig. 8). Despite the slight cold bias of the WRF simulation, we take the WRF-derived values as the best inputs available.

**3.2.4  Surface and material parameters**

Solving the USM energy balance equations requires a number of surface (albedo, emissivity, roughness length, thermal conductivity, and capacity of the skin layer) and volume (thermal capacity and volumetric thermal conductivity) material input
* * *
[6]http://medard-online.cz/

[Figure]

**Figure 7.** Comparison of WRF time series of temperature (left) and wind speed (right) to values measured at Prague, Kbely station. Shaded areas mark the time of the observation campaign.

[Figure]

**Figure 8.** Comparison of WRF vertical profiles of absolute temperature ($T$) and potential temperature ($\theta$) to values measured at Prague, Libuš station on 3 July 06:00 UTC.

[revised manuscript text omitted]

Figure 12 shows a south facing wall in the west arm of the west-east street measured from location 3. We evaluated the model performance in four points. All points are assigned the same material category (plastered brick wall, see Tables S2 and S3). Points 1, 2 and 3 are on a surface with the same colour, which is represented by an albedo of 0.2 in the model. Point 4 is placed on a surface of lighter colour (albedo of 0.7). The lighter surface colour in point 4 results in a significantly lower

5    measured peak temperature of 6–9 °C less than in other points. The model correctly captures the lower temperature in the point 4, although the modelled maximum in point 4 is a bit higher than the measured maximum. The effect of different albedos can be seen in Fig. 16 for points 2, 3 and 4, too. The observation that the model overestimates values at some evaluation points located in the lowest parts of the buildings can also be made at other observation locations (see location 4, Fig. 13, point 1 or location 5, Fig. S5, point 1).

10    In Fig. 12, daytime temperatures of points 1 and 2 are captured quite well, while the model overestimates the temperature in point 3. In reality, this point is shaded by an alcove until 08:10 UTC (see the IR picture in Fig. S10) and thus it is irradiated approximately 1 hour later than point 1. As a consequence, the increase of its temperature is delayed and the reached maximum temperature is 4 °C lower than in point 1. This facade unevenness is not resolved by the topography model in PALM and it thus predicts the same values for points 1 and 3.

15    Figure 13 shows the same comparison for a west facing wall in the south arm of the north-south street measured from location 4. The temperature course in point 7 demonstrates the effect of tree shading. The point is shaded from 13:10 to 14:50 UTC by the nearby tree. This results in a decrease of measured temperature between 13 and 15 UTC, whereas the temperature of other points (e.g. point 6) is increasing. The effect of the tree shading 
[revised manuscript text omitted]

**List of all relevant changes**

The manuscript went through major revision and it was substantially rewritten. Major changes are as follows. Based on reviewers' suggestions we have adjusted model setup and rerun the presented simulations. The text of all chapters is considerably revised. We added some new sections (Sect. 3.4.2 Sensitivity to material parameters, Sect. 4 Discussion). The text in Sect. 3.3 Evaluation of PALM-USM has been rewritten from scratch. We markedly extended the description of observation site and measurement methods (Sect. 3.1) and model setup description (Sect. 3.2). The aim of the changes was to clarify insufficient descriptions as identified by reviewers and better discussion of limitations of the current version of the model. Also, the Conclusions section was extended and new supplement file with additional figures and tables was prepared.

---

## Author Response (AR2)

We would like to thank both reviewers for the evaluation of our revised manuscript. Please, find below our responses. Reviewer comments are in italics and authors responses are in blue standard font. The revised manuscript with marked-up changes is attached at the end of this document.

Report #1
Submitted on 25 Aug 2017
Anonymous Referee #3

*The authors present a comprehensive study on coupling a new urban surface model with a large-eddy simulation model. Importance of the work is well justified in the introduction section. Model development and field measurement are described in details. Evaluation of model performance and sensitivity analysis is thorough. Overall this is a well written paper with high-quality presentation. I think the manuscript is ready for publication in its present form.*

Report #2
Submitted on 28 Aug 2017
Anonymous Referee #2

*I re-reviewed this paper and I thin that it improved considerably as compared to the paper that I was originally submitted. In its current form, the paper describes material that I consider to be a major contribution to geoscientific model development and thus fits well into the journal geoscientific model development. However, I think that the paper is quite lengthy (27 figures!) and for better readibility I suggest that the authors seek to shorten their paper. In my opinion, many parts of the WRF validation can be relegated to the supporting material. Furthermore, I think that part of the figures belonging to the sensitiviy study and the computational aspects could be summarized in a table, while both sections can in my opinion be shortened as well. Futhermore, I have a few remaining issues. My overall advice is to accept the paper after minor revisions.*

The revised paper is quite long. The originally submitted version has 26 pages, the revised text is 40 pages long. However, the extension of the text was necessary to fully address all issues raised by the reviewers. We do not consider it reasonable to do any substantial structural changes of the text in this stage as it could introduce new issues, e.g. inconsistencies in the text, unbalancing of parts or loss of clarity. We accepted the suggestion regarding the WRF part and we moved Figs. 6-8 to supplements. We also combined the Figs. 21-23 on the pages 28-29 (Sect. 3.4.2 Sensitivity to material parameters) into one figure (Fig. 18) what saved length of one page.

Major issues:

*1. I agree with the authors that the use of Monin-Obukhoc Similatiry Theory (MOST) is common practice in many surface parameterizations. In contrast to the study under scope in these studies, the lowest atmospheric model layer or the reference layer in case of an offline model, is usually above the surface and located in the surface layer or the upper part of the Roughness Sublayer. The authors apply a vertical grid spacing of only 2 m, which means that exchange between surface elements and the atmosphere is resolved on very small*

*scales. I question whether MOST still applies on these scales, and exchange between surface elements and the adjacent atmospheric layers should not be better simulated using a more detailed model including a 'law of the wall' approach.*

The reviewer points out correctly that, strictly, MOST is only applicable in the inertial sublayer, but not in the roughness sublayer below (see Basu & Lacser, 2017, BLM for a discussion on this matter). It is commonly accepted, that the atmospheric level to be used to evaluate MOST should fulfill the requirement z > 50 *z0, where z0 is the roughness length of the surface. In our case, the roughness lengths did not exceed values of 0.01 m, so that z should be larger than 0.5 m. The atmospheric grid level used for evaluating MOST in our study was taken 1 m away from the surface (0.5 * grid spacing) and thus fulfills this requirement and lies well within the inertial sublayer. The reviewer now mentions that a "law of the wall" approach should be included; but to our understanding, the law-of-the-wall is nothing more than the neutral limit of MOST. MOST can be regarded as a generalization of the law-of-the-wall taking into account the stability of the atmosphere. Now, stability is always regarded in terms of vertical stratification, so it can't be used for vertical surfaces. An alternative approach is thus needed to calculate the drag coefficient for heat. We do this by employing the formulation of Krayenhoff & Voogt (2007). Note, however, that the calculation of the friction velocity always follows MOST. For horizontal surfaces, stability correction is included, while for vertical surfaces, neutral stability is used, so that here, the law-of-the-wall scaling is actually applied.
We added a sentence on the calculation and the law-of-the-wall in the model description to satisfy the comment of the reviewer (P6, L2-5).

*2. The authors verify the developed model using measurements taken on 2 days in only one city. In my opinion this time period is very limitied and I suggest that the authors at least provide a discussion on how representative their results are for the performance of their model in other time periods, seasons, and other cities.*

The focus of this first version is on the simulation of heat wave episodes in the fully urbanised areas. This intention is stated in the text (abstract, Sect. 1.1, Sect. 2.). The set of implemented processes was chosen accordingly and also the validation and gathered observations correspond to this intention. We suppose that the model can be successfully used for the simulations of the urbanised areas in similar meteorological conditions. The limitations of the current version and the model setup are discussed thoroughly (mainly Sect. 4.1 and 4.2).
The addressing of these limitations and implementing of omitted processes is work in progress. This will extend the possible utilization of the model for other conditions. Of course, we suppose to perform validations for different areas, seasons and meteorological conditions once the enhanced version of the model is ready.
We added corresponding text to the Sect. 4.2 (P31, L12-18).

*3. The authors use an observed temperature profile to initialize their model, whereas they use WRF output for large-scale forcing. I wonder whether this does not lead to inconsistencies as possibly, the large-scale forcing is not equilibrium with the observed profiles. Why did the authors not consider to use the validated WRF profile to initialise their model?*

The WRF output is used only for the calculation of the large-scale advection tendencies and for nudging values in the upper layers of the free atmosphere in the present version of the simulations. We used this output also for the calculation of the initial profile in our original simulations but we decided to replace it by observed values as we consider the observation

as the best available information about the state of the free atmosphere. The question arises whether this approach is consistent.

We have shown in the WRF validation section (Sect. 3.2.3) that the modelled temperature profiles are in good agreement with radiosonde observations in higher layers of the atmosphere. The possible differences of the profiles are located in near surface layers. According to our sensitivity studies (not included in this paper), the near surface part of the initial profile has negligible influence on the results. The influence of the near surface initial temperatures is overlapped by the strong forcing from the surface during morning hours. We thus consider the choice whether to use observed or WRF initial profile to be marginal after the model spin-up time.

Minor points:

*1. Page 2, line 1: 'As more than a half ...' should be 'As more than half'*

Accepted

*2. Page2 , line 6/7: Please add that the UHI is manifests itself mainly during the evening and early night,*

The remark was added into the sentence. Note, that this phenomenon is also mentioned in the chapter 3.3 Evaluation of PALM-USM (P17, L27-28).

*3. Page 2, line 19: 'approaches to studying' should be 'approaches for studying'*

Accepted

*4. Page 4, line 15 to 28: very long sentence*

The paragraph has been rephrased and the long sentence split.

*5. Page 5, line 21: The energy budget doens not contain the latent heat flux. Please, explain here why*

The explanation and the reference to the discussion were added.

*6. Page 11, line 6/7 and page 13 line 27: latitue and longitude coordinates are given with 8 digits. Is such detail really necessary?*

One degree represents approx. 111 km in latitude and 72 km in longitude in the studied area. Thus the 1 m precision needs 5 decimal places which means 7 digits in total. We reduced the number of decimal places accordingly.

*7. Page 13, line 34: 'in 640 m' should be 'at 640 m'*

Accepted

*8. Page 19: line 13/14: My idea is that the deeper morning Atmospheric Boundary Layer (ABL)s in the morning over urban areas are not caused by the UHI, buth that both the UHI and the deeper ABLS stem from the same mechanism, namely the observation that urban areas are able to deliver extra heat to the air layers close to the surface and in the ABL.*

The sentence was reformulated to better describe the causes and consequences (P18, L5-7).

*9. Page 19, line 15: add 'potential' before 'temperature'*

Accepted

*10. Page 24, lines 15 to 20: I think these sentences are quite hard to follow. Probably, these sentences should be replaced by clearer sentences.*

The formulations were cleared (P23, L6-11).

**References**

Basu, S. and Lacser, A..: A Cautionary Note on the Use of Monin–Obukhov Similarity Theory in Very High-Resolution Large-Eddy Simulations, *Bound.-Lay. Meteorol.*, 163(2), 351–355, 2017.

[revised manuscript text omitted]